# Interneuron-specific signaling evokes distinctive somatostatin-mediated responses in adult cortical astrocytes

Letizia Mariotti[1,2], Gabriele Losi[1,2], Annamaria Lia[1,2], Marcello Melone[3,4], Angela Chiavegato[2], Marta Gómez-Gonzalo[1,2], Michele Sessolo[1,2], Serena Bovetti[5], Angelo Forli[5], Micaela Zonta[1,2], Linda Maria Requie[1,2], Iacopo Marcon[1,2], Arianna Pugliese[3], Cécile Viollet[6], Bernhard Bettler[7], Tommaso Fellin [5], Fiorenzo Conti[3,4,8] & Giorgio Carmignoto [1,2]

The signaling diversity of GABAergic interneurons to post-synaptic neurons is crucial to generate the functional heterogeneity that characterizes brain circuits. Whether this diversity applies to other brain cells, such as the glial cells astrocytes, remains unexplored. Using optogenetics and two-photon functional imaging in the adult mouse neocortex, we here reveal that parvalbumin- and somatostatin-expressing interneurons, two key interneuron classes in the brain, differentially signal to astrocytes inducing weak and robust GABA$_B$ receptor-mediated Ca$^{2+}$ elevations, respectively. Furthermore, the astrocyte response depresses upon parvalbumin interneuron repetitive stimulations and potentiates upon somatostatin interneuron repetitive stimulations, revealing a distinguished astrocyte plasticity. Remarkably, the potentiated response crucially depends on the neuropeptide somatostatin, released by somatostatin interneurons, which activates somatostatin receptors at astrocytic processes. Our study unveils, in the living brain, a hitherto unidentified signaling specificity between interneuron subtypes and astrocytes opening a new perspective into the role of astrocytes as non-neuronal components of inhibitory circuits.

[1] Neuroscience Institute, National Research Council (CNR), 35121 Padova, Italy. [2] Department of Biomedical Sciences, Università degli Studi di Padova, 35121 Padova, Italy. [3] Department of Experimental and Clinical Medicine, Università Politecnica delle Marche, 60126 Ancona, Italy. [4] Center for Neurobiology of Aging, INRCA IRCCS, 60121 Ancona, Italy. [5] Optical Approches to Brain Function Laboratory, Department of Neuroscience and Brain Technologies, Istituto Italiano di Tecnologia, 16163 Genova, Italy. [6] Inserm UMR894, Center for Psychiatry and Neuroscience, Université Paris-Descartes, 75014 Paris, France. [7] Departement of Biomedicine, University of Basel, 4031 Basel, Switzerland. [8] Foundation for Molecular Medicine, Università Politecnica delle Marche, 60126 Ancona, Italy. Letizia Mariotti and Gabriele Losi contributed equally to this work. Correspondence and requests for materials should be addressed to G.C. (email: giorgio.carmignoto@bio.unipd.it)

nhibition is a fundamental operational mechanism in the brain that is governed by GABAergic interneurons[1,2]. A large diversity of interneurons in terms of morphology, connectivity, molecular and functional properties ensures a signaling specificity to surrounding neurons. These unique features allow the different GABAergic interneurons to strictly control local network excitability and modulate synaptic transmission[1,2]. Among key interneurons in the neocortex are parvalbumin (PV)- and somatostatin (SST)-expressing interneurons. The former regulate the spike-timing and the gain of pyramidal neurons by targeting soma and proximal dendrites, while the latter control signal integration and synaptic plasticity by targeting the distal dendrites of pyramidal neurons[1–4].

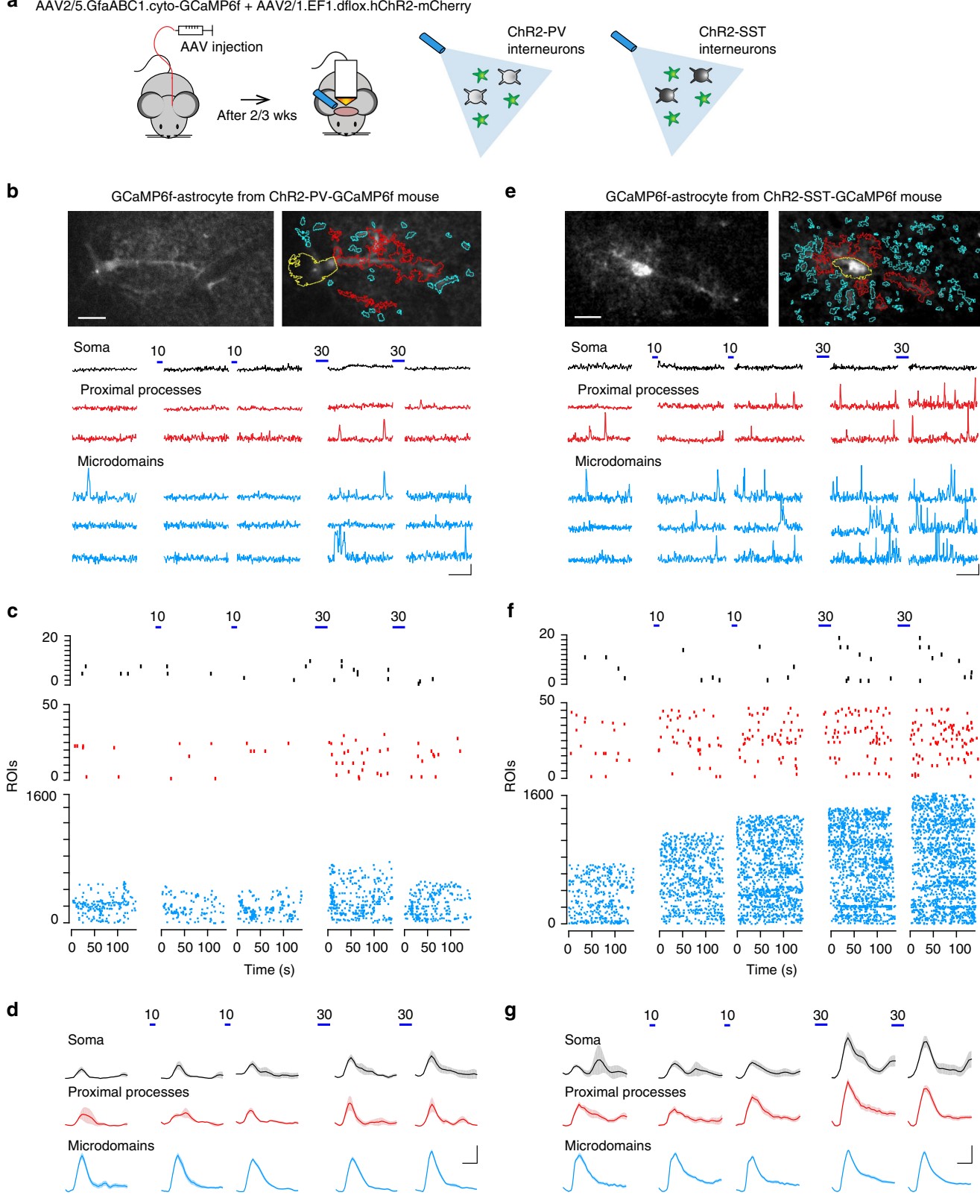

The glial cells astrocytes are additional modulatory elements of local network excitability and synaptic transmission[5,6]. In vivo studies revealed that different neurotransmitter systems, including glutamatergic, acetylcholinergic, and noradrenergic pathways[7–10], signal to astrocytes inducing in these cells complex cytosolic $Ca^{2+}$ changes that represent a key event in the action of astrocytes in local brain circuits[5,6,11–13]. Astrocytes have been proposed to crucially affect GABAergic synaptic transmission[14–16], but whether the various interneuron classes, which warrant the specificity of GABAergic signaling to postsynaptic neurons, also specifically signal to astrocytes is a question that remains completely unexplored. We here address this issue in the mouse somatosensory cortex (SSCx) and study the signaling to astrocytes of PV and SST interneurons by combining optogenetics with 2-photon $Ca^{2+}$ imaging in in vivo and in situ slice experiments.

## Results

**Experimental set-up**. To selectively stimulate PV or SST interneurons and evaluate potential $Ca^{2+}$ responses of astrocytes, we used an adeno-associated virus (AAV)-based strategy (Fig. 1a). Through this approach, we induced in PV interneurons of adult PV-Cre mice or SST interneurons of adult SST-Cre mice, the selective expression of the light-gated cation channel channelrhodopsin-2 (ChR2)[17] (Supplementary Fig. 1a–d) and in astrocytes the sparse expression of GCaMP6f (Supplementary Fig. 2a, b), a genetically encoded $Ca^{2+}$ indicator (GECI)[18–20] (ChR2-PV-GCaMP6f or ChR2-SST-GCaMP6f mice, see Methods).

**Parvalbumin interneurons evoke depressing $Ca^{2+}$ responses**. We first studied $Ca^{2+}$ signal dynamics in different compartments of GCaMP6f-expressing astrocytes including the soma, the proximal processes and the fine processes exhibiting spatially restricted $Ca^{2+}$ transients, i.e., $Ca^{2+}$ microdomains[19,21,22], in the SSCx of ChR2-PV-GCaMP6f mice (Fig. 1) and quantified their basic properties (Fig. 2). We applied 10 light pulses ($\lambda = 473$ nm, 150 ms duration, 1 Hz) that induced in ChR2-expressing PV interneurons firing activity (mean firing rate, $40.8 \pm 5.75$ Hz; Supplementary Fig. 3) comparable to that exhibited by these interneurons in awake mice[3]. No $Ca^{2+}$ elevations were observed in GCaMP6f-astrocytes either during (Supplementary Fig. 4a, b) or after (Fig. 1b, c) the 10 pulse stimulation. Equally ineffective was a second episode of this type of PV interneuron stimulation applied with a 5-min interval. Only a more prolonged activation by 30 pulses induced at both proximal processes and microdomains, an increase in the mean number of active sites (regions of interest, ROIs) and the frequency of $Ca^{2+}$ transients, whereas $Ca^{2+}$ event amplitude was increased in the proximal processes and remained unchanged in the microdomains (Fig. 1b–d; Fig. 2b, c, open bars, Supplementary Movie 1). As revealed by both the raster plots reporting the $Ca^{2+}$ events from all the monitored GCaMP6f-astrocytes (Fig. 1c) and the quantification of $Ca^{2+}$ response properties (Fig. 2), a second 30 pulse stimulation evoked reduced $Ca^{2+}$ elevations suggesting a depression of the

astrocyte response to successive episodes of PV interneuron activity.

**Somatostatin interneurons evoke potentiating $Ca^{2+}$ responses**. We then studied the $Ca^{2+}$ dynamics in GCaMP6f-expressing astrocytes from ChR2-SST-GCaMP6f mice. Optogenetic stimulation by 10 light pulses (150 ms duration, 1 Hz) induced in ChR2-expressing SST interneurons in vivo a firing activity (mean firing rate, $13.4 \pm 2.4$ Hz; Supplementary Fig. 3) comparable to that exhibited by these interneurons in awake mice[23,24]. To support this finding, we performed fluorescence-guided juxtasomal recordings in layer 2 of the somatosensory cortex of awake head-fixed mice, trained to remain still under the microscope (Supplementary Fig. 5a). Compatibly with previous studies, SST interneurons exhibited periods of spontaneous firing containing brief bursts of high instantaneous activity (Supplementary Fig. 5b–d) similar to that induced in these neurons by our optogenetic stimulation. In contrast to PV interneurons, 10 light pulse stimulation of SST interneurons was sufficient to activate GCaMP6f-astrocytes (Fig. 1e, f) inducing an increase in the mean number of active ROIs and the mean frequency of $Ca^{2+}$ events in both proximal processes and microdomains (Fig. 2b, c, closed bars; Supplementary Movie 2). As in the astrocyte response to PV stimulation, the amplitude of $Ca^{2+}$ events increased only in the proximal processes. These data indicate that astrocytes are more sensitive to SST than PV interneuron activity. Interestingly, with respect to the first, a second episode of SST interneuron activation by 10 pulses applied with a 5-min interval induced a greater $Ca^{2+}$ response in astrocytes and a similar potentiation was observed following the two successive 30 pulse stimulations. The raster plots (Fig. 1f) and the quantification of $Ca^{2+}$ response in the proximal processes and microdomains (Fig. 2b, c) confirm that, with respect to the first, a second episode of SST interneuron stimulation (by either 10 or 30 pulses) was significantly more effective, rather than less effective as in the case of PV interneuron stimulation, indicating a potentiation of the astrocyte response to SST interneuron signaling. The response depression to PV interneurons and the response potentiation to SST interneurons are confirmed by the significant leftward and rightward shift, respectively, in the cumulative distributions of $Ca^{2+}$ event frequency (Fig. 2d). These in vivo results were fully replicated in SSCx slices obtained from young ChR2-PV- and ChR2-SST-GCaMP6f mice (Supplementary Figs. 6 and 7).

**Integration of $Ca^{2+}$ microdomain responses**. With respect to spontaneous events, the mean amplitude of evoked $Ca^{2+}$ elevations in the proximal processes was significantly increased in response to activation of PV or SST interneurons (Fig. 1d, g, red traces and Fig. 2b, right panel), whereas that of evoked $Ca^{2+}$ microdomains was unchanged (Fig. 1d, g, blue traces; Fig. 2c, right panel; see also Supplementary Figs. 6 and 7). These data suggest that the interneuron signaling is essentially encoded into an increased microdomain frequency in the astrocytic fine processes and subsequently integrated in the proximal processes into larger amplitude $Ca^{2+}$ elevations. Whether this signal integration

**Fig. 1** Calcium signal dynamics reveal differential astrocyte responses to PV and SST interneuron activation. **a** Schematic of the in vivo experimental approach (left) and of the optogenetic stimulation of ChR2-PV or ChR2-SST interneurons (right). **b** Top, images of a representative GCaMP6f-astrocyte in layer 2/3 SSCx from an adult ChR2-PV-GCaMP6f mouse with the ROIs defined by GECIquant software for the $Ca^{2+}$ response to the first 30 light pulse stimulation (blue lines) of PV interneurons at the soma (yellow), proximal processes (red), and microdomains (blue), scale bar, 20 μm (see Supplementary Movie 3). Bottom, $Ca^{2+}$ signal dynamics at different astrocytic compartments before and after successive 10 and 30 light pulse PV interneuron activations. Scale bars, 50 s, 20% d$F/F_0$. **c**, **d** Raster plots of $Ca^{2+}$ peaks (**c**) and mean time course of $Ca^{2+}$ transients (**d**) from all in vivo monitored GCaMP6f-astrocytes, at rest and following PV interneuron stimulations. Scale bar, 5 s, 20% d$F/F_0$. **e–g** Same as in **b–d**, but for ChR2-SST-GCaMP6f mice and SST interneuron stimulation

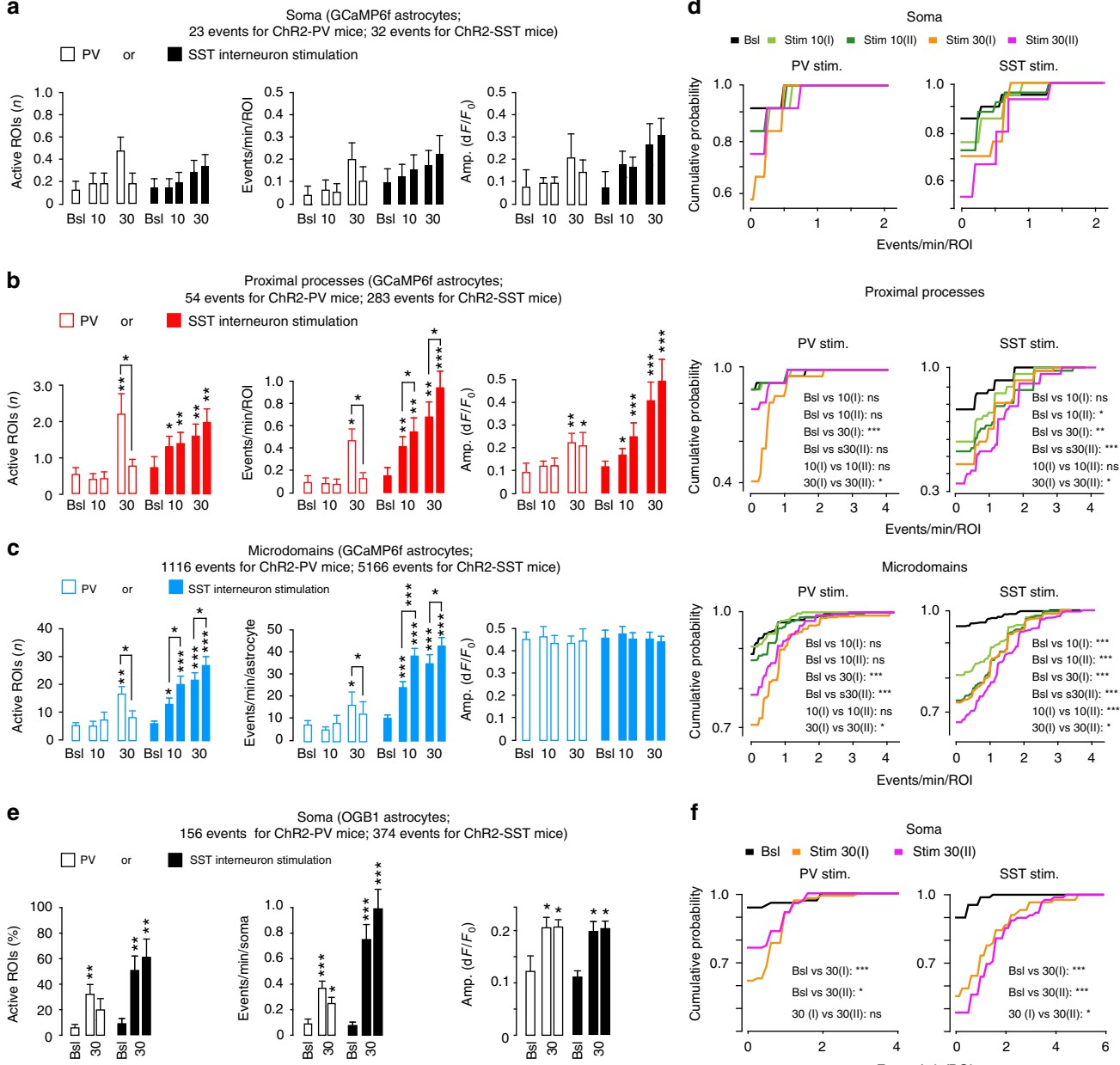

**Fig. 2** Properties of the astrocyte response to PV and SST interneurons in vivo. **a–c** Average data of $Ca^{2+}$ signal dynamics at different compartments of GCaMP6f-astrocytes from ChR2-PV- or ChR2-SST-GCaMP6f mice in response to PV interneuron (14 astrocytes, 7 mice) and SST interneuron signaling (20 astrocytes, 8 mice). **d**, **f** Cumulative distributions of astrocytic $Ca^{2+}$ events confirming significant response depression to successive PV interneuron stimulations and significant response potentiation to successive SST interneuron stimulations. $*p \leq 0.05$, $**p \leq 0.01$; $***p \leq 0.001$, Kolmogorov–Smirnov test. Exact $p$-values for the data reported in this as well as all the other figures are reported in the Supplementary Table 4e. Mean percentage of responsive astrocytes, mean frequency and amplitude of somatic $Ca^{2+}$ events in response to PV interneuron (white bars, 100 astrocytes, 6 ChR2-PV mice) or SST interneuron (black bars, 90 astrocytes, 7 ChR2-SST mice) optogenetic activation in the SSCx in vivo after loading with OGB-1 and the specific astrocytic marker SR101. Data are represented as mean ± SEM

involves also the astrocytic soma is, however, unclear. Indeed, due to the documented sparse nature of GCaMP6f expression in astrocytes[20], somatic $Ca^{2+}$ events in ChR2-PV- and ChR2-SST-GCaMP6f mice were analyzed from a limited number of cells. We therefore addressed this issue in ChR2-PV or ChR2-SST mice after loading a large number of astrocytes with chemical $Ca^{2+}$ indicators, such as Oregon Green BAPTA-1 or Fluo-4-AM, and SR101, a specific astrocytic marker[25]. By evaluating $Ca^{2+}$ signals in vivo (Fig. 2e; Supplementary Fig. 8) and in slice preparations (Supplementary Fig. 9), we observed somatic $Ca^{2+}$ response of astrocytes to both PV and SST interneurons. These results suggest

that the integration of microdomain $Ca^{2+}$ signals involve, in addition to the proximal processes, also the astrocytic soma. The cumulative distributions of event frequency also revealed a significant potentiation of somatic $Ca^{2+}$ signals in response to SST interneurons and a tendency to depression in response to PV interneurons (Fig. 2f; Supplementary Fig. 9f).

Altogether, these data demonstrate that astrocytes differently respond to PV and SST interneurons and change their $Ca^{2+}$ response as a function of the previous history of activity in the surrounding GABAergic interneuron type-specific network.

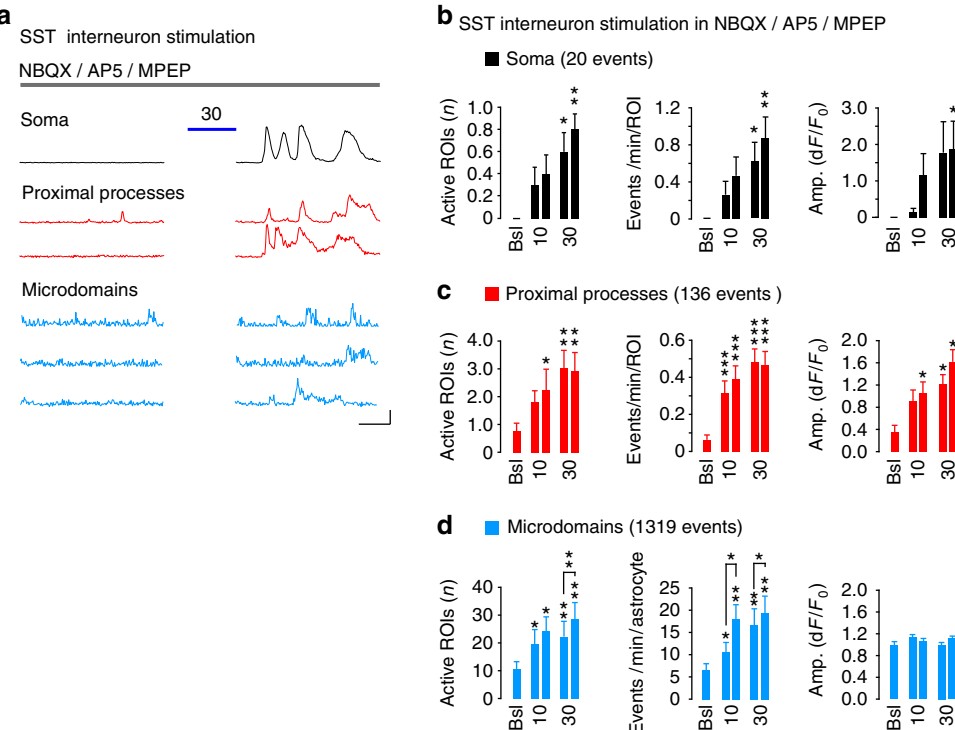

**Fig. 3** Astrocyte response to SST interneurons does not depend on glutamatergic transmission. **a** Representative traces and **b–d** quantitative evaluation of GCaMP6f-astrocyte responses to SST interneuron activation (10 or 30 light pulses) in SSCx slice preparations (9 astrocytes, 9 slices, 3 mice) in the presence of NBQX (10 μM), D-AP5 (50 μM), and MPEP (50 μM). Scale bars, 20 s, 50% d$F/F_0$. Data are means ± SEM

We next performed additional in vivo and brain slice experiments to obtain further insights into PV and SST interneuron signaling and to control the specificity of the astrocyte response. In SSCx slice preparations, we found that the increased $Ca^{2+}$ elevations induced by interneuron signaling in the different astrocytic compartments, including microdomains, were mediated by activation of $GABA_B$ receptors ($GABA_B$Rs) because they were abolished by the specific $GABA_B$R antagonist SCH50911 (Supplementary Figs. 7 and 9). These responses were independent on TRPA1 channel activation (Supplementary Fig. 10), a channel that has been previously proposed to modulate spontaneous microdomain activity[26]. The microdomain responses were detected with a delay of $14.93 \pm 1.34$ s from the onset of 10 pulse activation of SST interneurons and with longer delays in the proximal processes and soma, which may reflect slow intracellular $GABA_B$R-mediated signaling pathways. A similar delay ($14.97 \pm 1.30$ s; $p = 0.949$) of the microdomain response was measured following the onset of 30 pulse activation of SST interneurons (Supplementary Fig. 4c, d) suggesting that the duration of interneuron stimulation does not affect the delay of the astrocyte response. Altogether, these results suggest that astrocytes do not respond rapidly to synaptically released GABA and accompany a sustained interneuron activity with multiple, slowly developing $GABA_B$R-mediated $Ca^{2+}$ elevations.

We also evaluated the time window of the astrocyte response potentiation to SST interneuron signaling by increasing the interval between the first and the second SST interneuron stimulation. We found that the potentiation is a transient phenomenon as it was absent with 20-min intervals ($p = 0.688$) and observed with 10-min intervals only as a small, albeit significant ($p = 0.031$), increase in the mean number of active microdomains (Supplementary Fig. 11).

To rule out the possibility of unspecific effects produced on astrocytes by prolonged illumination with the imaging laser[27], we monitored $Ca^{2+}$ signals in astrocytes from ChR2-SST- or ChR2-PV-GCaMP6f mice without optogenetic stimulation. We failed to detect during the imaging sessions any significant change in the frequency or the amplitude of $Ca^{2+}$ peaks in the different astrocytic compartments, in both in vivo (Supplementary Fig. 12a–c) and SSCx slice experiments (Supplementary Fig. 12d–f).

The results obtained with the specific $GABA_B$R antagonist SCH50911 indicate a direct effect of synaptic GABA on astrocytes. The astrocyte response to interneurons might, however, be due, at least in part, to an increased local network excitability deriving from the inhibition exerted by SST interneurons on PV interneuron firing that, in turn, reduces the inhibition of PV interneurons to pyramidal neurons, ultimately enhancing glutamatergic signaling[28,29]. To address this hypothesis, we stimulated SST interneurons in the presence of different glutamate receptor selective blockers, i.e., NBQX (10 μM, 2,3-dihydroxy-6-nitro-7-sulfamoyl-benzo(F)quinoxaline) for AMPARs, D-AP5 (50 μM, D-2-amino-5-phosphonopentanoate) for NMDARs and MPEP (50 μM, Methyl-6-(phenylethynyl) pyridine) for the metabotropic glutamate type 5 receptor. Under these conditions, the overall $Ca^{2+}$ response of astrocytes to SST interneuron signaling was unchanged (Fig. 3) suggesting that glutamatergic signaling does not contribute to the astrocyte response to SST interneuron activity.

The depression of astrocytic $Ca^{2+}$ elevations in response to PV interneurons and the potentiation in response to SST interneurons could be due to a change in the synaptic release of GABA rather than to an intrinsic astrocytic property. To address this hypothesis, we measured the firing rate from PV and SST interneurons and the amplitude of evoked inhibitory postsynaptic currents (IPSCs) from pyramidal neurons, at each light pulse in the two sets of 10 and in the two sets of 30 light pulses applied with a 5-min interval. In the case of PV interneurons,

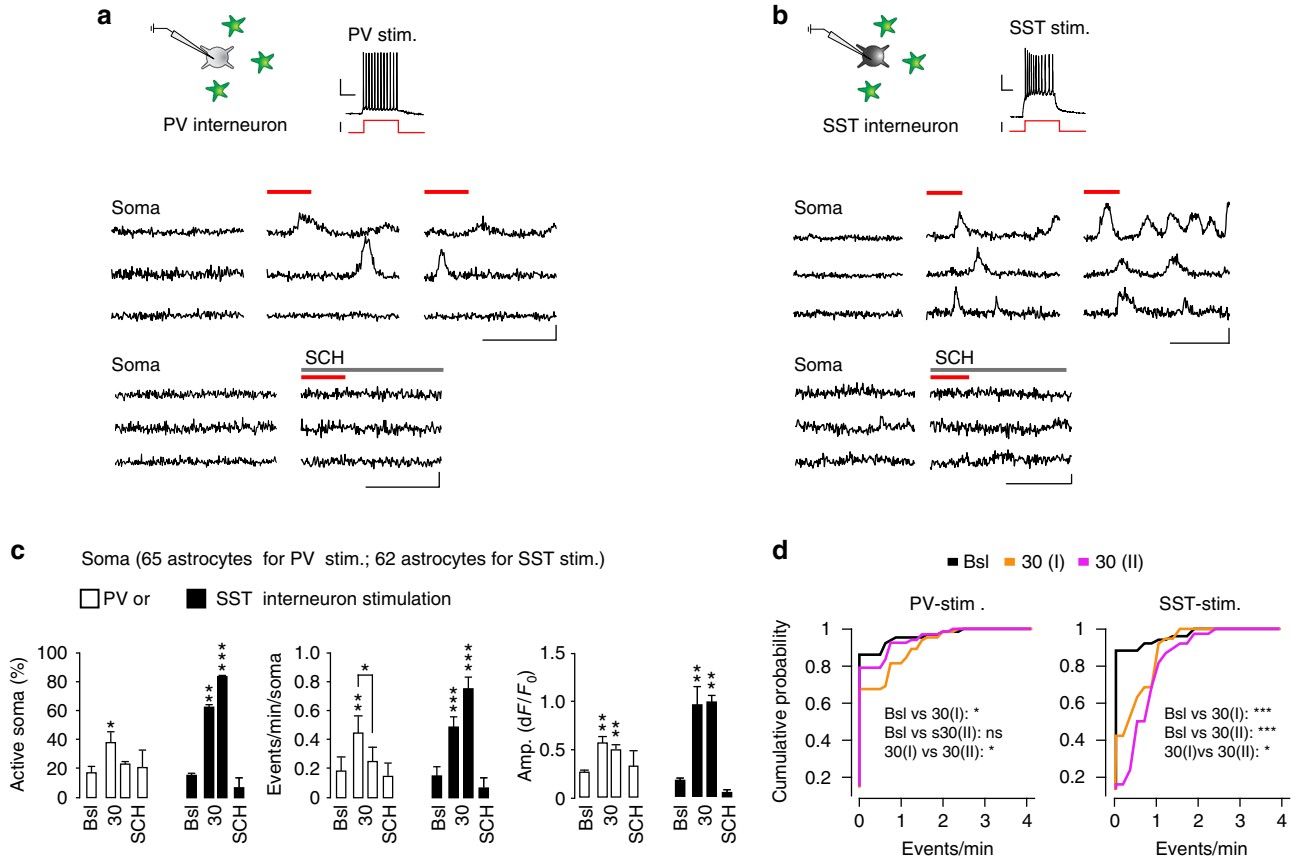

**Fig. 4** GABAergic signaling from individual PV or SST interneurons in SSCx slices is sufficient to recruit neighboring astrocytes. **a**, **b** Top, schematics of patch-clamp experiments and representative AP firing induced by intracellular depolarizing current pulse injections into a PV (mean firing rate, $11.8 \pm 2.1$ Hz) or a SST interneuron (mean firing rate $9.0 \pm 1.44$ Hz; burst firing rate, $30 \pm 4.8$ Hz). Scale bars, 100 ms, 20 mV, 200 pA. Bottom: somatic $Ca^{2+}$ signal dynamics from representative astrocytes before and after two sequences of 30 current pulses (red lines) delivered to PV or SST interneurons in absence or presence of SCH50911 (SCH, 50 μM). Astrocytes from an area within 100 μm from the patched interneuron were considered. Scale bars, 50 s, 20% d$F/F_0$. **c** Mean percentage of responsive astrocytes and mean $Ca^{2+}$ oscillation frequency in response to individual PV (65 astrocytes, 4 slices, 4 mice) or SST interneuron (62 astrocytes, 4 slices, 4 mice) stimulation. Data are represented as mean $\pm$ SEM. **d** Cumulative distributions of $Ca^{2+}$ event frequency after two subsequent 30 current pulse injections. With respect to the first stimulation, the astrocyte response to the second PV interneuron stimulation was significantly depressed, whereas that to the second SST interneuron stimulation was significantly potentiated (Kolmogorov–Smirnov test, *$p \leq 0.05$, **$p \leq$ 0.01, ***$p \leq 0.001$)

besides an unchanged firing rate during each set of light pulses, we observed a reduction in IPSC amplitude (Supplementary Fig. 13e) that can be indicative of a $GABA_A$ receptor desensitization[30] and/or a decrease in synaptic GABA release. The IPSC reduction during the two sets of 30 pulse PV interneuron stimulation was, however, similar, in terms of both its time course ($p = 0.66$ and $p = 0.83$ for the fast and the slow time decay component, respectively) and amplitude ($p = 0.55$; Supplementary Fig. 13e, f) suggesting that a decrease in GABA release cannot explain the impairment of the astrocyte response to the second PV interneuron stimulation. In the case of SST interneurons, both the firing rate (Supplementary Fig. 13g–i) and the evoked IPSC amplitude (Supplementary Fig. 13j–l) were unchanged suggesting that the astrocyte response potentiation to SST interneurons is unlikely due to an increase in the amount of synaptically released GABA. However, direct measurements of GABA concentrations would be necessary to validate this conclusion.

**Individual PV or SST interneurons recruit nearby astrocytes.**
The optogenetic activation induces synaptic GABA release from a large number of ChR2-expressing interneurons. We asked

whether a more restricted release of GABA, such as that deriving from activation of a single interneuron, also recruits neighboring astrocytes. To address this question, we used SSCx slices from tdTomato-floxed::PV- and SST-Cre mice as well as G42 and GIN mice expressing the enhanced green fluorescence protein (GFP) in a subset of PV- or SST-positive interneurons, respectively, after astrocyte loading with Fluo-4-AM and SR101. We found that astrocytes from an area within 100 μm from the patched interneuron were effectively recruited by individual PV or SST interneuron activation (30 depolarizing current pulses, 300-ms duration, 1 Hz) and they exhibited potentiated $Ca^{2+}$ elevations in response to SST interneuron and depressed $Ca^{2+}$ elevations in response to PV interneuron stimulation (Fig. 4d). Both responses were sensitive to the $GABA_BR$ antagonist SCH50911 (Fig. 4a–c). A localized synaptic GABA release from a single interneuron is, therefore, sufficient to recruit nearby astrocytes evoking a response with the same properties as those of the response observed in the optogenetic experiments.

**Crucial role of the neuropeptide somatostatin.** To clarify whether the mechanism of the higher sensitivity of astrocytes to SST than PV interneurons derives from a closer position of

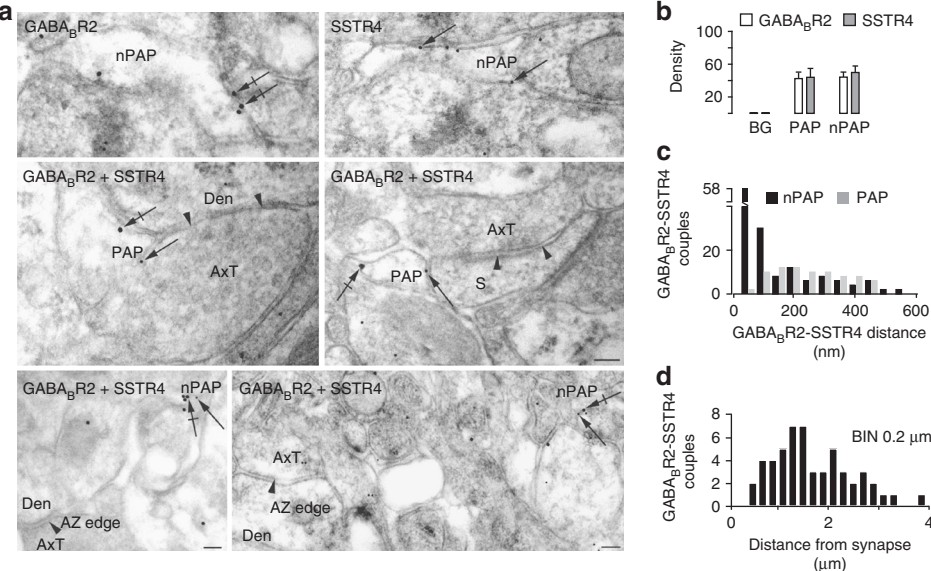

**Fig. 5** GABA$_B$Rs and SST4Rs colocalize at non perisynaptic astrocytic processes (nPAPs). **a** Upper row, GABA$_{B2}$ and SSTR4 immunogold EM single-labeled nPAPs (arrows point to 18 and 12 nm membrane-associated gold particles, respectively). Middle row, GABA$_{B2}$-SSTR4 double-labeled PAPs nearby to an axo-dendritic shaft (left) and an axo-spinous (right) symmetric synapse (arrowheads point to edges of active zones). Lower row, double-labeled nPAPs not in the proximity of symmetric synapses (arrowheads). **b** GABA$_{B2}$ and SSTR4 relative density (particles/$\mu m^2$) in double-labeled PAPs ($n = 40$) and nPAPs ($n = 77$) is higher than background (BG; $p < 0.0001$, Mann–Whitney test), whereas it is similar between PAPs and nPAPs. **c** Distance distribution of immunogold GABA$_{B2}$ and SSTR4 pairs at PAP membranes (40 pairs from 37 PAPs) and nPAPs (100 pairs from 97 nPAPs). **d** Lateral position of GABA$_{B2}$ and SSTR4 pairs with an edge-to-edge distance within 50 nm with respect to the closest AZ margin of a symmetric synapse (53 couples from 53 nPAPs). Axt, axon terminal; Den, dendrite; s, spine. Scale bars, 100 nm

astrocytic processes to SST than PV interneuron synapses, we performed electron microscope (EM) immunocytochemical experiments. Obtained data revealed, however, similar structural relationships between astrocytic processes and PV/SST interneuron synapses (Supplementary Fig. 14; Table 1).

The different astrocytic response to PV and SST interneurons may be due to a different molecular signaling between these interneuron classes and astrocytes. To address this hypothesis, we studied whether the neuropeptide SST, that is released in addition to GABA by SST interneurons[31], contributes to the response of astrocytes. In double-immunogold EM experiments, we first investigated whether astrocytes express SST receptors, focusing on the SST type 4 receptor (SSTR4), which was previously described in astrocytes from cell culture and hippocampal slice preparations[32,33]. Post-embedding EM experiments revealed that astrocytic processes express both GABA$_{B2}$ and SSTR4 with similar densities at perisynaptic astrocytic processes (PAPs) and at processes not contacting symmetric synapses (nPAPs; Fig. 5a, b). Most interestingly, pairs of GABA$_{B2}$-SSTR4 gold particles exhibiting an edge-to-edge separation distance within 50 nm, were found almost exclusively at nPAPs (Fig. 5c) suggesting functional interactions between the two receptors. Notably, the GABA$_{B2}$-SSTR4 couples (<50 nm) were found at nPAPs at a mean distance of $1.79 \pm 0.10$ $\mu m$ from symmetric synapses (Fig. 5d).

The specificity of the anti-GABA$_{B2}$ and anti-SSTR4 antibodies used in our EM immunocytochemical study was validated in experiments on GABA$_{B2}$[34] and SSTR4[35] knockout mice (Supplementary Figs. 15 and 16; Supplementary Tables 2 and 3).

We then asked whether activation of astrocytic SSTRs could induce per se Ca$^{2+}$ signal changes. We found that application of the neuropeptide SST (1–2 $\mu M$) induced a weak, albeit significant, increase of Ca$^{2+}$ event frequency in GCaMP6f-astrocytes from SSCx slices (Fig. 6a, b). In the presence of tetrodotoxin (TTX, 1 $\mu M$) and a cocktail of different neurotransmitter receptor

antagonists, such as SCH (50 $\mu M$, for GABA$_B$R), MPEP (50 $\mu M$, for mGluR5), and PPADS (100 $\mu M$, for P2YR), the response of astrocytes to SST was unchanged suggesting a direct action of SST on astrocytic SSTRs (Fig. 6c).

These observations prompted us to directly investigate whether a synergistic action of SST may occur on GABA-mediated Ca$^{2+}$ response of astrocytes to SST interneuron signaling. First, we found that in the presence of CYN 154806 (20 $\mu M$), a SSTR antagonist, SST interneuron activation by 10 light pulses was ineffective and only following 30 pulses were Ca$^{2+}$ changes observed (Fig. 6d; Supplementary Movie 3). Second, in the presence of CYN 154806, the astrocyte Ca$^{2+}$ response induced by a second episode of 30 pulse stimulation was not potentiated and it rather exhibited a significant reduction in the mean number of microdomains ($p = 0.007$) and the mean event frequency ($p = 0.003$, Fig. 6d, e, lower panels). Therefore, astrocytic Ca$^{2+}$ responses to SST interneurons in the presence of CYN 154806 become comparable to those evoked by PV interneurons. We next asked whether bath perfusion of the neuropeptide SST (1–2 $\mu M$) could result in an astrocyte response to PV interneurons similar to that induced by SST interneurons. We found that under these conditions, astrocytes did not exhibit the response depression upon the second episode of 30 pulse activation of PV interneurons, albeit they failed to respond to 10 pulse stimulation (Fig. 7a–d). It appears, therefore, that PV interneuron activation coupled with exogenous SST peptide can mimic, at least in part, the astrocyte response to SST interneurons.

The results reported above suggest an important role of the neuropeptide SST in the astrocyte response to SST interneurons. To evaluate the degree of astrocytic response facilitation or depression, we calculated the mean ratio of the second to the first response of the astrocytes to the two successive activations of SST interneurons (RR, see Methods), comprehensive of both Ca$^{2+}$ elevations at proximal processes and microdomains, in the absence or presence of the SSTR blocker, and of PV interneurons

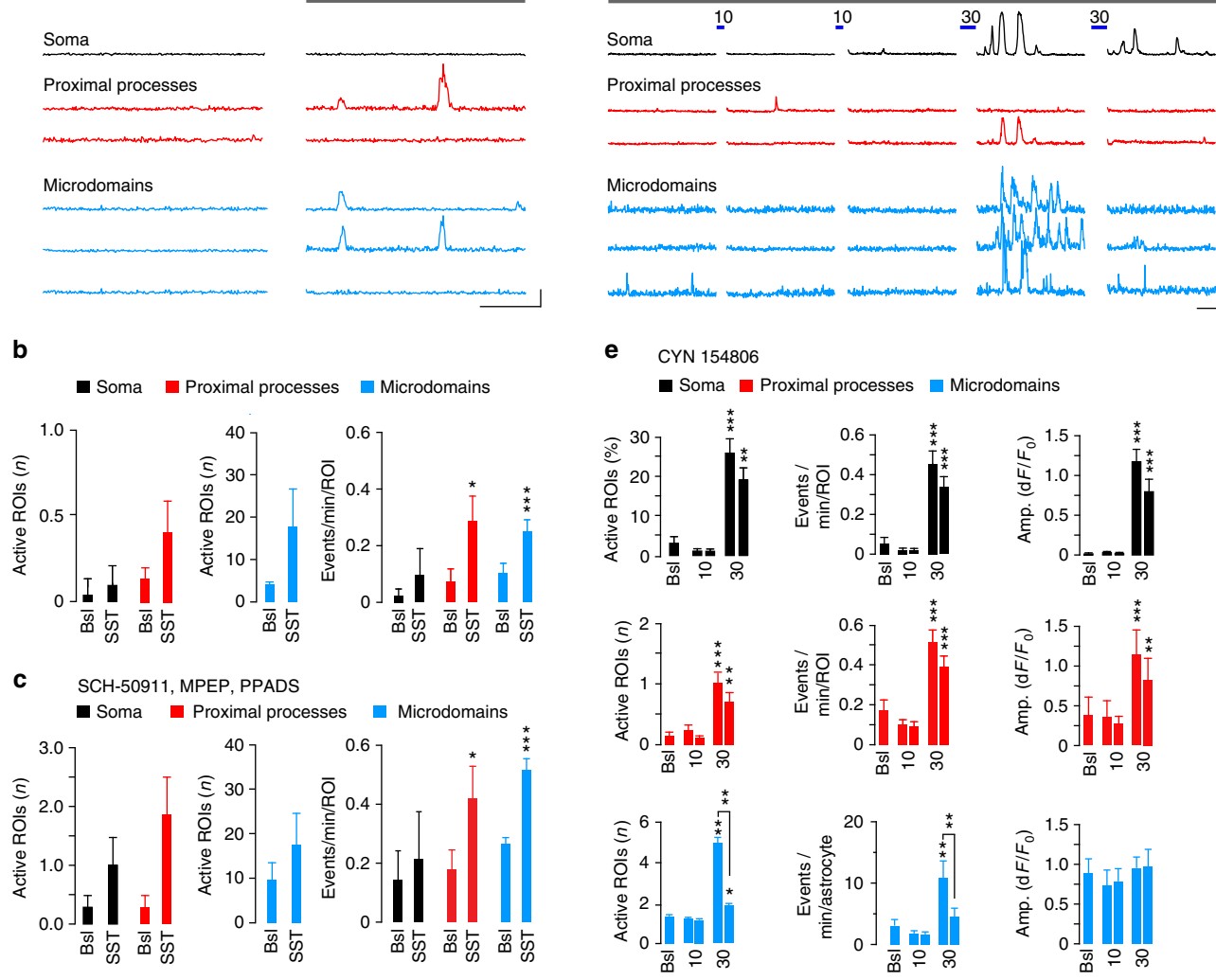

**Fig. 6** SST interneuron signaling specificity to astrocytes depends on the neuropeptide somastostatin. **a** Representative traces and **b**, **c** quantitative evaluation of GCaMP6f-astrocyte responses to the neuropeptide SST in SSCx slices in the absence (28 astrocytes, 8 slices, 3 mice) or the presence (9 astrocytes, 9 slices, 3 mice) of antagonists of GABA$_B$ (SCH50911, 50 μM), mGlu5 (MPEP, 50 μM) and purinergic (PPADS, 100 μM) receptor antagonists and in the continuous presence of TTX (1 μM). Scale bars, 50 s, 20% d$F/F_O$. The response of GCaMP6f-astrocytes is not affected by the mix of receptor antagonists. **d** Representative Ca$^{2+}$ traces of a GCaMP6f-astrocyte before and after subsequent 10 and 30 pulse SST interneuron activations in the presence of 20 μM CYN 154806. Scale bars, 50 s, 20% d$F/F_O$. **e** Mean number of active ROIs, event frequency and amplitude at soma, proximal processes and microdomains of GCaMP6f-astrocytes (57 astrocytes, 10 slices, 5 mice). Data are represented as mean ± SEM

in the presence or absence of the neuropeptide SST (Fig. 8a). Obtained values confirmed that the astrocyte response was significantly depressed upon the 30 pulse successive stimulations of PV interneurons and it was significantly facilitated upon both 10 and 30 pulse successive stimulations of SST interneurons. When the activation of SSTRs was prevented, the astrocyte response to SST interneuron stimulations was depressed and became undistinguishable from that to PV interneurons. Furthermore, when the PV interneuron stimulations were applied in the presence of the neuropepetide SST, no depression of the astrocyte response was observed (Fig. 8a).

## Discussion

We analyzed the GABAergic signaling to astrocytes of PV- and SST-expressing interneurons in the mouse somatosensory cortex in vivo and in situ. Our work provides evidence for the following main findings. First, astrocytes are more sensitive to SST than PV interneuron signaling. Second, astrocytic Ca$^{2+}$ responses weaken

or strengthen upon successive episodes of activity in PV and SST interneuron circuit, respectively. Third, both the high sensitivity and the potentiated Ca$^{2+}$ response to SST interneurons crucially depend on the neuropeptide Somastostatin, released by these interneurons, and the following activation of SST receptors expressed at astrocytic processes in close association with GABA$_B$ receptors.

The higher sensitivity of astrocytes to SST than PV interneuron signaling was revealed by the observation that a prolonged activity in the PV interneuron circuit was necessary to evoke astrocytic GABA$_B$R-mediated Ca$^{2+}$ elevations, whereas a short episode of activity in the SST interneuron circuit was sufficient to activate astrocytes. These results provide indication of a signaling specificity to the astrocytic network of GABAergic interneuron subtypes. As a further support to this, we observed that astrocytes activated by a first episode of activity in the PV interneuron circuit, either failed to respond or exhibited a response depression upon a successive episode of similar activity, both in terms of frequency and amplitude of evoked Ca$^{2+}$ elevations. In contrast,

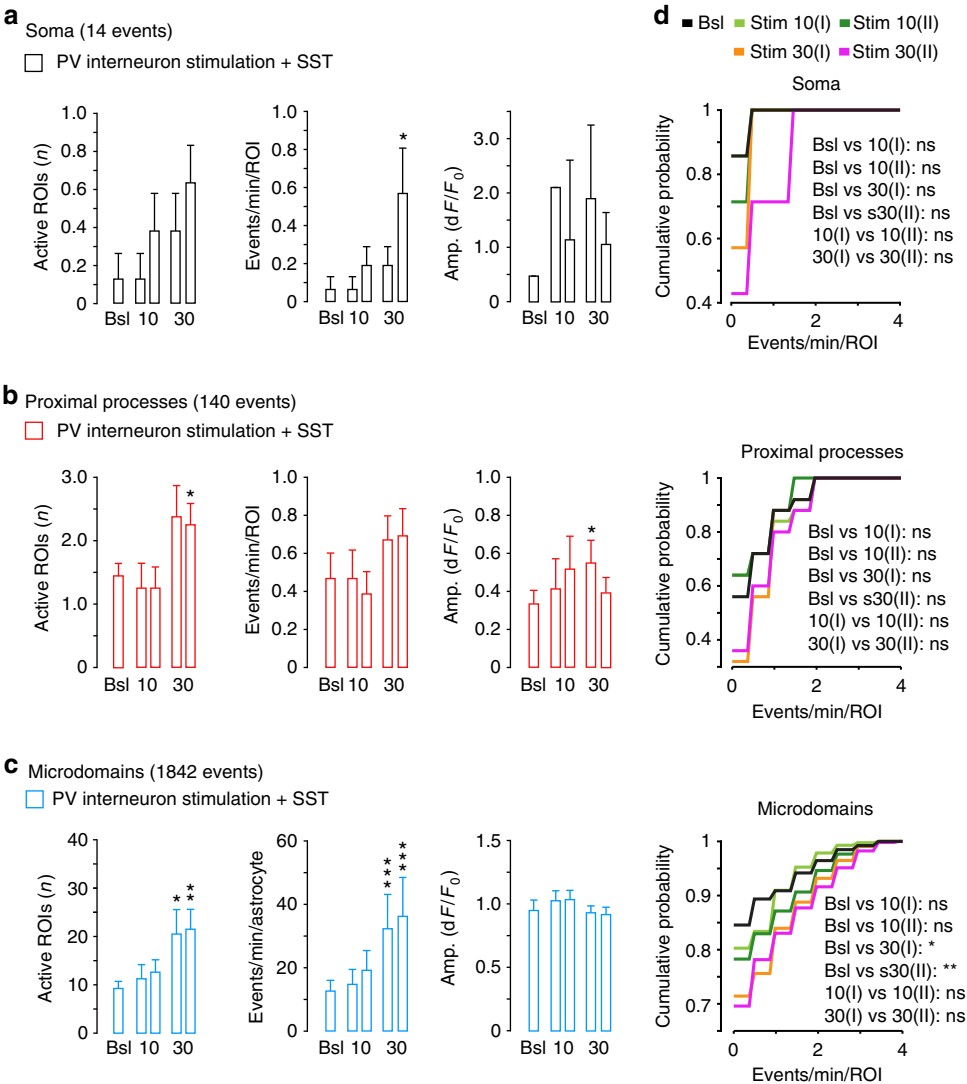

**Fig. 7** The neuropeptide somatostatin prevents the depression of the astrocyte response to PV interneurons. **a–c** Average data of GCaMP6f-astrocyte Ca²⁺ response to PV interneuron activation (10 or 30 light pulses) from SSCx slice preparations (8 astrocytes, 8 slices, 3 mice) in the presence of somatostatin (1–2 µM). **d** Cumulative distributions of Ca²⁺ events show no response depression to successive 30 pulse stimulations of PV interneurons (for proximal processes, $p = 0.694$; for microdomains, $p = 0.957$, Kolmogorov–Smirnov test)

the robust long-lasting astrocytic Ca²⁺ oscillations evoked by an initial SST interneuron activation were further strengthened by successive episodes of activity and almost all astrocytes from the SSCx layer 2–3 were ultimately recruited indicating that a sustained activity in the SST interneuron circuit is complemented by a sustained activity in the astrocytic network. Notably, evoked GABA_BR-mediated Ca²⁺ elevations in astrocytes weaken or strengthen when surrounding PV or SST interneurons are repetitively activated demonstrating a distinguished plasticity of the astrocyte response to these two interneuron subtypes.

Clues for a modulation of the astrocytic Ca²⁺ elevations evoked by neuronal signals have been previously obtained in cell cultures[36] and slice preparations[37–40], but totally unexplored was this property of astrocytes in the real functioning in vivo context. Our study unveils that astrocytes in the adult mouse somatosensory cortex modulate their Ca²⁺ responses as a function of previous states of activity in the surrounding interneuronal network suggesting the existence in these glial cells of a form of cellular memory. This remarkable plasticity of the astrocyte response and the signaling specificity of GABAergic interneuron subtypes suggest that astrocytes are functionally associated to

inhibitory circuits. Accordingly, the sustained recruitment of a large astrocytic network to SST interneuron circuit, coupled with the typically slow neuromodulatory action of gliotransmission[5], may contribute to the homeostatic regulation of dendritic inputs and signal integration in pyramidal neurons that is a primary role of SST interneurons[4,23,41–43]. Consistent with this hypothesis, GABA_BR-mediated Ca²⁺ elevations in astrocytes has been recently reported to evoke a release of the gliotransmitter glutamate[44] that significantly potentiates synaptic transmission in the hippocampus[45].

The differential properties of the astrocyte response to PV and SST interneurons were observed following both optogenetic light pulses, that activate a large number of ChR2-expressing interneurons, and intracellular current pulse activation of individual interneurons. A synaptic GABA release from a single interneuron is, therefore, sufficient to recruit neighboring astrocytes indicating that each interneuron is in extensive and efficient functional contacts with the surrounding astrocytic network.

The mechanism governing the dynamics of Ca²⁺ microdomains is poorly defined. We here report that Ca²⁺ elevations evoked by synaptic GABA in different astrocytic compartments,

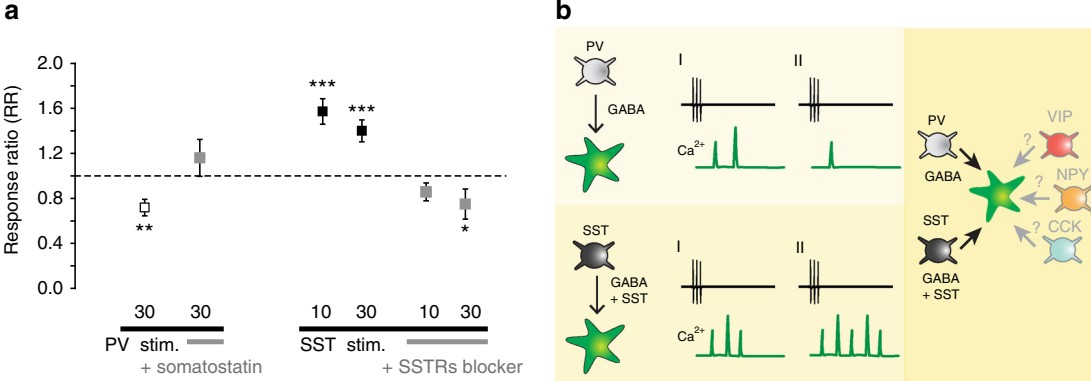

**Fig. 8** Specificity of neuropeptide-releasing interneuron signaling to astrocytes. **a** Astrocyte response ratio (RR, see Methods) of the second versus the first 10 or 30 light pulse stimulation of PV or SST interneurons in the absence or presence of the neuropeptide SST or of a somatostatin receptors (SSTRs) blocker. $*p \leq 0.05$, $**p \leq 0.01$, $***p \leq 0.001$. Data are represented as mean ± SEM. **b** Schematic of the signaling specificity to astrocytes of SST and PV interneurons (left) and of the potential recruitment of the astrocytic network by different neuropeptide-releasing interneurons (right)

including microdomains, were mediated by activation of the $GABA_BR$ that previous studies reported to be functionally expressed in astrocytes[44–49]. We also evaluated the possible contribution of $Ca^{2+}$ influx through TRPA1 channels that a recent study using plasma membrane-targeted GCaMP3 proposed to modulate spontaneous microdomain activity[26]. In our study, we found that both spontaneous and evoked microdomains were not affected in the presence of the TRPA1 antagonist HC030031 (Supplementary Fig. 8). These findings should, however, be interpreted with caution as cytosolic GCaMP6f signal may understate near plasma membrane $Ca^{2+}$ microdomain activity. The signaling pathways mediating the $Ca^{2+}$ microdomain activity remain, therefore, to be clarified. The contribution of IP3 signaling pathway in the $Ca^{2+}$ elevations induced by PV and SST interneuron signaling also remains to be defined. Future analyses on $IP_3R2^{-/-}$ mice expressing ChR2 on PV or SST interneurons and GCaMP6f on astrocytes will help clarifying this important issue. It will be also important to validate the GABAergic interneuron type-specific response of astrocytes in awake non-anesthetized animals, given that astrocyte $Ca^{2+}$ signaling can be affected by anesthetics[50,51].

The unique properties of the astrocyte response to SST interneurons, i.e., the high sensitivity to synaptic GABA and the potentiated $Ca^{2+}$ elevations, were mediated by the neuropeptide somatostatin that is co-released with GABA by SST, but not PV interneurons[31]. Consistently, the astrocyte response was observed after an intense activation of Somatostatin interneurons that triggers somatostatin release[52,53]. As a further support to this signaling mode of SST interneurons, our immunogold experiments revealed a close association between SSTRs and $GABA_BRs$ at astrocytic processes that is consistent with functional interactions occurring between the two receptors. $GABA_BR$-SSTR4 couples (<50 nm) were fundamentally found at processes not contacting symmetric synapses, rather than at perisynaptic astrocytic processes suggesting that once released from SST interneurons, GABA and SST travel a certain distance before reaching $GABA_B$-SSTR4R couples. Therefore, a possible synergistic action between the two astrocytic receptors, which may account for the enhanced $Ca^{2+}$ elevations in response to SST interneurons, would eventually occur with a certain delay after intense firing in these interneurons, as we observed in our experiments. All together, our results suggest that astrocytes do not respond rapidly to synaptically released GABA. Rather, they sense GABA tonic elevations and accompany a sustained interneuron activity with multiple, slowly developing $Ca^{2+}$ elevations.

The mechanism of the $Ca^{2+}$ response depression induced by PV interneuron signaling to astrocytes remains to be defined. Desensitization and internalization of $GABA_B$ receptors have been reported to occur in neurons[30,54], but whether similar mechanisms also occur in astrocytes and account for the reduction of the astrocytic response to PV interneurons is unknown. If this were the case, a prevention of $GABA_B$ receptor desensitization or internalization by a concomitant activation of astrocytic SST and $GABA_B$ receptors might account for the powerful recruitment of astrocytes by SST interneuron signaling. Consistent with this hypothesis, we found that exogenous neuropeptide SST applications prevented the depression of the astrocyte response, otherwise, occurring after the second episode of PV interneuron stimulation, whereas it was not sufficient to induce a response potentiation. A low SST concentration, possibly due to poor penetration of the neuropeptide into the brain tissue, might account for such a partial effect.

Subtypes of SSTRs have been reported to interact and form heterodimers with other G-protein-coupled receptors generating receptor oligomers that can have different desensitization and internalization properties as well as unique pharmacological profiles[33,55–58]. In support of this view, SSTR activation in cultured astrocytes has been reported to synergistically potentiate the $Ca^{2+}$ response to α1 adrenergic receptor-mediated signaling[55].

Different SSTRs are also expressed on neuronal axon terminals and they are proposed to cooperate with pre-synaptic GABARs in the control of glutamate release[31]. We cannot, therefore, exclude that a change in the network activity mediated by neuronal SSTR activation may contribute to modulate the astrocyte response to SST interneuron signaling.

Besides SST, various neuropeptides, such as neuropeptide Y (NPY), vasointestinal polypeptide (VIP), colecystokinin (CCK), neurokinin B and enkephalins, are synthesized with GABA by distinct interneuron classes and their release has been proposed to contribute to interneuron-specific actions by targeting neuronal receptors[31]. Our results suggest that neuropeptides are also used by interneurons to target astrocytic receptors. Besides SSTRs, astrocytes express, indeed, different receptors for neuropeptides that are released by interneurons including NPY, VIP, CCK, and opioids[31,33,55,59]. Based on these observations, we postulate that the mode of the SST interneuron signaling that we describe here may represent a general mechanism in brain networks by which neuropeptide-releasing interneurons recruit to their specific circuits neighboring astrocytes inducing $Ca^{2+}$ elevations with unique spatial-temporal properties (Fig. 8b).

In the somatosensory cortex of adult mice, we identified a novel signaling mode of GABAergic interneurons that revealed the presence of cell-specific interneuronal-astrocytic networks. Our study opens a new perspective into the role of astrocytes as distinct functional components of interneuron type-specific neocortical circuits.

## Methods

**Mouse strains and adeno-associated virus injections.** We used C57BL/6 wild type (WT) mice and the following transgenic mice: Tg(GadGFP)45704Swn (GIN), (CB6-Tg(Gad1-EGFP)G42Zjh/J) (G42), Pvalb<tm1(cre)Arbr>(PV-Cre) and Sst < tm2.1(cre)Zjh>(SST-Cre), and tdTomato reporter line B6;129S6-Gt(ROSA) 26Sor$^{tm14(CAG-tdTomato)Hze}$/J. All procedures were conducted in accordance with the Italian and European Community Council Directive on Animal Care and approved by the Italian Ministry of Health. Injections of viral vectors AAV2/1.EF1.dflox.hChR2(H134R)-mCherry.WPRE.hGH (Penn Vector Core, Addgene 20297) or AAV1.EF1a.DIO.hChR2(H134R)-eYFP.WPRE.hGH, Addgene 20298), carrying the doublefloxed ChR2 sequence, and AAV5.GfaABC1DcytoGCaMP6f.SV40, carrying the astrocytic promoter GfaABC1D, which induces a sparse expression of the $Ca^{2+}$ indicator GCaMP6f in astrocytes, were performed into the SSCx of postnatal day 35–50 (P35-P50) PV-Cre or SST-Cre mice anesthetized with Zoletil (30 mg/kg) and Xylazine (20 mg/kg). Depth of anesthesia was assured by monitoring respiration rate, eyelid reflex, vibrissae movements, and reactions to pinching the tail and toe. Injections of the two viral vectors were performed after drilling one or two holes (0.5 mm dia) into the skull over the SSCx at a distance of 1.5 mm (1.5 µl to each hole, 0–1.5 mm posterior to Bregma, 1.5 mm lateral to sagittal sinus, and 150 µm depth) using a pulled glass pipette in conjunction with a custom-made pressure injection system. After injections, the skin was sutured and mice were revitalized under a heat lamp and returned to their cage. Optogenetic and imaging in vivo experiments were performed in P50-P65 mice, 2 weeks after injections. For slice experiments, PV-Cre and SST-Cre pups (P0–P2) anesthetized by hypothermia and secured into a modeled platform were injected. Optogenetic and imaging experiments were carried out on SSCx slices from P15-P25 mice.

**Slice preparation, dye loading and patch-clamp recordings.** Coronal SSCx slices of 350 µm were obtained from mice at postnatal days P15-25. Animals were anesthetized as reported above, the brain removed and transferred into an ice-cold solution (ACSF, in mM: 125 NaCl, 2.5 KCl, 2 CaCl₂, 1 MgCl₂, 25 glucose, pH 7.4 with 95% O₂, and 5% CO₂). Slices were cut in the solution reported in Dugue et al.[60] and then kept for 1 min in the solution (in mM): 225 D-mannitol, 2.5 KCl, 1.25 NaH₂PO₄, 26 NaHCO₃, 25 glucose, 0.8 CaCl₂, 8 MgCl₂, 2 kynurenic acid with 95% O₂, and 5% CO₂. Finally, slices were kept in ACSF at 30 °C for 20 min and then maintained between 19 and 22 °C for the entire experiment. In a set of experiments, SSCx slices were incubated with the $Ca^{2+}$ sensitive dye Fluo-4 AM (10 µM; Life Technologies) and the selective astrocyte dye Sulforhodamine 101[25] (SR101, 0.2 µM, Sigma Aldrich, Italy), as previously described[44]. For the in vivo experiments, the bulk loading of cortical astrocytes was performed with the dye OGB-1 AM (final concentration 1 mM, Thermo Fisher Scientific, USA) and SR101 (final concentration 0.5 mM). For whole-cell patch-clamp recordings, slices were perfused in a submerged chamber at a rate of 3–4 ml/min with (in mM): 120 NaCl, 2.5 KCl, 1 NaH₂PO₄, 26 NaHCO₃, 1 MgCl₂, 2 CaCl₂, 10 glucose, pH 7.4 (with 95% O₂ and 5% CO₂). Neurons were visualized under a confocal microscope (TCS-SP5-RS, Leica Microsystems, Germany) or a Multiphoton Imaging System (Scientifica Ltd, UK) equipped with a CCD camera for differential interference contrast (DIC) image acquisition. Single-cell recordings were performed in voltage- or current-clamp configuration using a multiclamp 700B amplifier (Molecular Devices, USA). Signals were filtered at 1 kHz and sampled at 10 kHz with a Digidata 1440 s interface and pClamp 10 software (Molecular Devices). The pipette resistance was 3–4 MΩ. Access resistance was monitored throughout the recordings and was between 8.4 and 24.2 MΩ. Neurons that had a >15% change in access resistance were discarded. Whole-cell intracellular pipette solution was (in mM): 145 K-gluconate, 5 MgCl₂, 0.5 EGTA, 2 Na₂ATP, 0.2 Na₂GTP, 10 HEPES, to pH 7.2 with KOH, osmolarity, 280 ÷ 290 mOsm. Data were not corrected for the liquid junction potential. Recordings were analyzed with Clampfit 10.3.

**Drug applications.** Drugs applied to the slice perfusion solution were: SCH50911 (20–50 µM), CYN 154806 (20 µM), and somatostatin (SST, 1–2 µM) from Tocris (UK), Tetrodotoxin (1 µM), CGP52432 (20 µM), HC030031 (80 µM), NBQX (10 µM), APV (50 µM), MPEP (50 µM), PPADS (100 µM) from Abcam (UK).

**Ca²⁺ imaging.** To image $Ca^{2+}$ dynamics in GCaMP6f-astrocytes, we used 2-photon laser scanning microscope in both in vivo (Ultima IV, Bruker) and brain slice (Multiphoton Imaging System, Scientifica Ltd., UK) preparations equipped with a pulsed red laser (Chameleon Ultra 2, Coherent, USA) tuned at 920 nm. Power at sample was controlled in the range 5–10 mW. The excitation wavelengths used were 920 nm for GCaMP6f, 740 nm for Fluo-4 and 830 nm for OGB-1. SR101 is visible at both 740 and 920 nm. Images were acquired with a water-immersion lens (Olympus, LUMPlan FI/IR 20×, 1.05 NA), with a field of view between 700 × 700

µm and 120 × 120 µm at 1–3.5 Hz acquisition frame rate. Each $Ca^{2+}$ signal recording was performed in cortical layers 2–3 for about 2 min and 30 s with 5 min interval between the first and the second stimulation of interneurons by 10 or 30 light pulses, whereas a longer interval of about 10 min was applied before initiating the first 30 pulse stimulation. In control experiments, the same imaging protocol was applied without optogenetic stimulation. A confocal laser scanning microscope (TCS-SP5-RS, Leica Microsystems, Germany) equipped with two lasers tuned at 488 nm and 543 nm was used only in a subset of experiments to continuously monitor $Ca^{2+}$ signals from GCaMP6f-astrocytes during optogenetic light activation of PV or SST interneurons in SSCx slices (Supplementary Fig. 4), an unfeasible condition in our 2-photon experiments. In in vivo imaging experiments, P55-P65 ChR2-PV-GCaMP6f or ChR2-SST-GCaMP6f mice were anesthetized with urethane (20% urethane, ethylcarbamate; SIGMA Aldrich). Animal pinch withdrawal and eyelid reflex were tested to assay the depth of anesthesia. Dexamethasone sodium phosphate (2 mg/kg body weight) was injected intramuscularly to reduce cortical stress response during surgery and prevent cerebral edema. Atropine (0.05 mg/kg body weight) was injected subcutaneously to avoid saliva accumulation. Both eyes were covered with an eye ointment to prevent corneal desiccation during the experiment. We monitored the respiration rate, heart rate, and core body temperature throughout the experiment. The mouse was head-fixed and a craniotomy of 2–3 mm in diameter was drilled over the SSCx. Mice were mounted under the microscope with a metal head-post glued to the skull. Imaging was performed through a water-immersion lens (Olympus, LUMPlan FL/N 20×, 1.05 NA) at a resolution of 512 × 512 pixels with zoom 4, leading to a field of 50.7 × 50.7 µm in superficial layers (50–150 µm below the cortical surface) and acquired at 1–2 Hz. Imaging session lasted up to 2 h.

**Optogenetic stimulation.** Full-field photo-stimulation of ChR2-expressing interneurons consisted of 150 ms light pulses ($\lambda = 473$ nm) delivered by a blue module laser diode (MLD, COBOLT, Solna, SE), which was collimated and coupled under the objective with an optic fiber (ThorLabs, NJ, USA) held at 26° angle above the brain tissue. The optic fiber was 300 µm in diameter with a 0.22 NA. The resulting illuminated ellipse was 550 µm long and 150 µm wide.

**Two-photon-targeted juxtasomal recordings in vivo.** Experimental procedures followed what previously described[61]. In brief, for recordings in anesthetized mice, PV-Cre/tdTomato and SST-Cre/tdTomato double-transgenic mice were injected at P0 with AV2/1.EF1.dflox.hChR2(H134R)-mCherry.WPRE.hGH (Addgene 20297). Four to five weeks after virus injections mice were anesthetized with urethane (2 g/kg) and a small craniotomy (~1 mm × 1 mm) was opened onto the mouse skull. The patch pipette (resistance: 4–9 MΩ) was filled with ACSF solution mixed with Alexa Fluor-488 (20 µM, Invitrogen Thermo Fisher, USA) and lowered to cortical layer 2/3 (110–300 µm from the brain surface). tdTomato-positive neurons were targeted by imaging their fluorescence with the two-photon microscope ($\lambda = 920$ nm or 730 nm). Full-field optogenetic stimulation of interneurons was performed as in Zucca et al.[61] Light intensity was 0.2–6 mW at the fiber tip. For recordings in non-anesthetized mice, 2 weeks before the experiment mice were anesthetized with 2% isoflurane/0.8% oxygen and a custom metal plate was mounted with dental cement on the skull. Habituation sessions were performed on each day (starting 2–3 days after plate implantation) with a gradually increasing duration (from 15 to 60 min, for 7–10 days). The day of the recording, mice were anesthetized with isofluorane and a small craniotomy was opened on the somatosensory cortex as described above. After the surgery, mice recovered for at least 30 min before the beginning of the experimental session. Electrical signals were amplified by a Multiclamp 700B, low-pass filtered at 10 kHz, digitized at 50 kHz with a Digidata 1440 and acquired with pClamp 10 (Axon instruments, USA). Electrophysiological traces were analyzed using Clampfit 10 software.

**Pre-embedding electron microscopy.** Three C57BL/6 (P21) mice were anesthetized with chloral hydrate (12% i.p., 300 mg/kg) and perfused through the ascending aorta with physiological saline solution followed by a mixture of 4% paraformaldehyde (PFA) and 0.2% glutaraldheyde in PBS. Brains were post-fixed in the same fixative used for perfusion for 3 days and parietal cortex was cut serially in the coronal plane (40 µm sections) with a vibratome and immediately processed for immunoperoxidase according to previous pre-embedding electron microscopy protocols[62]. For antibody specificity on the SSTR4, two 9-month-old mice (WT and SSTR4 KO[35]) were perfused through the ascending aorta with a flush of physiological saline solution followed by 4% PFA in PBS. Brains were post-fixed in the same fixative for 1 h, cryopreserved and then frozen until cutting by a vibratome. For antibody specificity on the GABA$_{B2}$ two 7-week-old WT BALB/c JR1 mice and two GABA$_{B2}$ KO BALB/c JR1 mice[34] were perfused through the ascending aorta with physiological saline solution followed by 4% PFA in PBS. Brains were post-fixed in the same fixative used for perfusion for 7 days until cutting by a vibratome. For PV and SST visualization sections were incubated in a solution containing rabbit polyclonal anti-PV (1:500; raised against rat muscle PV; PV28, RRID:AB_10013386, Swant, Switzerland) or rat monoclonal anti-SST primary antibodies (1:80; raised against synthetic 1–14 cyclic SST, MAB 354; RRID: AB_2255365 EMD Millipore, Germany; 2 h at room temperature and overnight at 4 °C)[63]. The following day, sections were incubated in a solution containing the

appropriate biotinylated secondary antibodies (1:200; Jackson ImmunoResearch, USA; 1 h at room temperature). Antibody bindings sites were visualized by avidin–biotin peroxidase complex, 3,3 diaminobenzidinetetrahydrochloride and $H_2O_2$[62]. Method specificity was verified by substituting primary antibodies with phosphate buffer (PB) or non immune serum. Subsequently, embedding procedure of immunoperoxidase processed sections was performed as described[62]. Small blocks of embedded tissue containing layers 2/3 of the SSCx were selected, glued to blank epoxy and sectioned with an ultramicrotome (MTX; Research and Manu-facturing Company Inc., USA). The most superficial ultrathin sections (60 nm) were collected and mounted on 200 mesh copper grids, stained with Sato's lead and examined with a Philips EM 208 and CM10 electron microscope (Eindhoven, The Netherlands) coupled to a MegaView-II high resolution CCD camera (Soft Imaging System, Germany). Identification of labeled and unlabeled profiles was based on established morphological criteria[64]. Microscopic fields were selected and captured at original magnifications of 30,000 or ×50,000. According to the different post-synaptic targets of PV and SST interneurons, PV interneuron immunopositive terminals were sampled at axo-somatic, proximal axo-dendritic, and axo-axonic synapses, and SST interneuron immunopositive terminals at axo-dendritic shaft and axo-spinous synapses PAPs were then identified and quantification at sym-metric synapses of PV and SST interneurons performed.

**Post-embedding electron microscopy.** Three C57BL/6 (P21) were anesthetized with chloral hydrate (12% i.p.; 300 mg/kg) and perfused through the ascending aorta with a flush of physiological saline solution followed by 4% PFA in PBS. Brains were post-fixed in the same fixative for 7 days and parietal cortex was cut serially in the coronal plane in 50 μm thick sections with a vibratome. Sections were processed for an osmium-free embedding method[62,65,66]. Chips including layers 2/3 of SSCx, were selected, glued to blank resin blocks and sectioned with an ultramicrotome. Thin sections (60–80 nm) were cut and mounted on 300 mesh nickel grids and processed for immunogold post-embedding labeling[62,65,66]. For $GABA_{B2}$ and SSTR4 visualization, grids were incubated overnight (26 °C) in a solution containing anti-$GABA_{B2}$ mouse monoclonal antibody (1:50; raised against amino acids 183–482 mapping within an extracellular domain of $GABA_{B2}$ of human origin, specific for detection of $GABA_{B2}$ of mouse, rat and human; H10; sc-393270, Santa Cruz Biotechnology Inc., USA) and anti-SSTR4 rabbit polyclonal antibody (1:50; raised against amino acids 171–220 of SSTR4 of human origin, specific for detection of SSTR4 of mouse, rat and human; H50; sc-25678, RRID: AB_2196360, Santa Cruz Biotechnology), and then incubated for 2 h (26 °C) in a solution containing anti-mouse and anti-rabbit secondary antibodies conjugated to 18 and 12 nm gold particles (1:20; 115–215–068, 111–205–144, Jackson Immu-noResearch, USA). Grids were finally stained with uranyl acetate and Sato's lead. The optimal concentration of antibodies to $GABA_{B2}$ and SSTR4 was sought by testing several dilutions; the concentration yielding the lowest level of background labeling and still immunopositive elements was used to perform the final studies. Gold particles were not detected when primary antiserum was omitted. When normal serum was substituted for immune serum, sparse and scattered gold par-ticles were observed, but they did not show any specific relationship to subcellular compartments. Ultrathin sections (15 ultrathin sections/animal) were examined at ×50,000–85,000 and fields that included at least 1 immunolabeled astrocytic profile and/or perisynaptic astrocytic process associated with a symmetric synapse exhi-biting a clear pre-synaptic (AZ) and post-synaptic specialization were selected[64,67]. For determining the relative density of $GABA_{B2}$ and SSTR4 double-labeled astrocytic profiles, pyramidal cell nuclei were also identified: gold particles within labeled structures counted and areas calculated using ImageJ (NIH, Bethesda, MD, USA). Background was calculated by estimating labeling density over pyramidal cell nuclei ($0.54 \pm 0.05$, $n = 12$ for $GABA_{B2}$ and $0.57 \pm 0.02$, $n = 12$ for SSTR4)[62,68]. Particle densities were counted in perisynaptic (PAPs; $42.43 \pm 7.27$ for $GABA_{B2}$ and $44.02 \pm 10.66$, $n = 40$ for SSTR4) and non perisynaptic astrocytic processes (nPAPs; $44.26 \pm 6.07$ for $GABA_{B2}$ and $49.88 \pm 7.87$ for SSTR4) and compared with background labeling. Gold particles were considered associated with plasma membrane if they were within 15 nm of the extracellular side of the membrane, and cytoplasmic if they were 25 nm from the extracellular processes. Edge-to-edge separation distance between $GABA_{B2}$ and SSTR4 membrane-associated gold par-ticle pairs were measured and the distribution of the separation distance between immunogold labeled $GABA_{B2}$ and SSTR4 pairs was determined[69,70]. In astrocytic processes, pairs of immunogold labeled $GABA_{B2}$ and SSTR4 with an edge to edge distance within 50 nm, were also localized with respect to the closest AZ margin of symmetric synapses. Lateral position of a pair was defined as the distance along the plasma membrane from the AZ edge to the middle point between the two particles, and measured using ImageJ. For experiments in KO mice and relative controls, microscopical fields containing spines, axon terminals and astrocytic processes with at least one gold particle for $GABA_{B2}$ analysis, proximal and distal dendrites, axon terminals, and astrocytic processes for SSTR4 analysis were randomly selected. For both pre- and post-embedding studies, all material from WT and KO mice was processed in parallel. Acquisition of ultramicroscopical fields and density analysis of WT and KO mice were performed in a blind manner.

**Immohistochemistry and cell counting.** For the evaluation of the number of GCaMP6f-expressing astrocytes and neurons we prepared 100 μm thick brain slices from young and adult animals injected with AAV-ChR2 and AAV2/5.GfaABC.

cyto.GCaMP6. Slices were fixed in cold 4% PFA for 2 h, washed with PBS and processed for double immunofluorescence staining. First, we incubated floating sections for 1 h in the Blocking Serum (BS: 1% BSA, 2% goat serum and 1% horse serum in PBS) and 0.2% TritonX-100. We then performed a second incubation with primary antibodies mixed and diluted in BS and 0,02% TritonX-100 (16 h at 4 °C). Primary antibodies used were: anti-NeuN antibody (RRID:AB_2298772, 1:400 mouse, Millipore MAB377) plus anti-GFP (RRID:AB_221477, 1:200 rabbit, Invi-trogen Thermo-Scientific, A21311), and anti-glial fibrillary acidic protein (GFAP, RRID:AB_10013382, 1:300 rabbit, Dako, Denmark, Z0334) plus anti-GFP (RRID: AB_221568, 1:200 mouse, Invitrogen Thermo-Scientific, A11120). The anti-GFP antibodies were used to enhance the GCaMP6f fluorescence. After washing with PBS, slices were incubated for 2 h at room temperature with secondary antibodies conjugated with Alexa Fluor-488 (for staining GFP) and with Alexa Fluor-633 (for staining NeuN or GFAP; Invitrogen Thermo-Scientific, 1:500). Slices were then washed and mounted on glass coverslips. Negative controls were performed in the absence of the primary antibodies. For the evaluation of the number of PV- and SST-interneurons expressing ChR2, PV-cre/tdTomato and SST-cre/tdTomato double-transgenic mice were injected at P0 with AAV1.EF1a.DIO.hChR2(H134R)-eYFP.WPRE.hGH (see above for details). Four weeks after virus injection, mice were anesthetized with urethane (2 g/kg) and perfused transcardially with 0.9% saline solution, followed by 4% PFA in 0.1 M PB, pH 7.4. Brains were post-fixed for 6 h, cryoprotected in a 30% sucrose solution in 0.1 M PB pH 7.4 and frozen. Free-floating coronal serial sections (40 μm) from injected PV-cre/tdTomato and SST-cre/tdTomato mice were collected and stained against parvalbumin or somatos-tatin, respectively. The following primary antibodies were used: anti-parvalbumin (RRID: AB_477329, 1:1000 mouse, Sigma P3088) and anti-somatostatin (RRID: AB_2255365, 1:200 rat, Millipore MAB 354). Secondary antibodies consisted of: goat anti-mouse 647 (RRID: AB_141725, 1:800, Molecular Probes A21236) and goat anti-rat 647 (RRID:AB_141778, 1:800, Molecular Probes A21247). Sections were mounted on SuperFrost slides (Molecular Probes), air dried, and coverslipped in polyvinyl alcohol with diazabicyclo-octane (DABCO). Confocal image z-stacks were captured through the thickness of the slice at 1 μm steps and used for double-labeled cell count using an open source ImageJ plugin.

**Data analysis.** Detection of astrocyte ROI containing $Ca^{2+}$ elevations was per-formed with ImageJ in a semi-automated manner using the GECIquant plugin[20]. The software was used to identify ROIs corresponding first to the soma (>30 μm²; confirmed by visual inspection), then to the proximal processes (>20 μm² and not corresponding to the soma) and finally to the microdomains (between 1 and 20 μm² corresponding to neither the soma nor the proximal processes). All pixels within each ROI were averaged to give a single time course F(t). Analysis of $Ca^{2+}$ signals was performed with ImageJ (NIH) and a custom software developed in MATLAB 7.6.0 R2008 A (Mathworks, Natick, MA, USA). To compare relative changes in fluorescence between different cells, we expressed the $Ca^{2+}$ signal for each ROI as $dF/F_0 = (F(t) − F_0)/(F_0)$. We then defined as *baseline trace* for each ROI the points of the $Ca^{2+}$ trace with absolute values smaller than twice the standard deviation of the overall signal. Significant $Ca^{2+}$ events were then selected with a supervised algorithm as follows. Firstly, a new standard deviation was calculated on the *baseline trace*, and all local maxima with absolute values exceeding twice this new standard deviation were identified. Secondly, of these events, we considered significant only those associated with local calcium dynamics with amplitude larger than threefold the new standard deviation. The amplitude of each $Ca^{2+}$ event was measured from the 20th percentile of the fluorescent trace interposed between its maximum and the previous significant one (see Supplementary Fig. 17). Essentially, this procedure combines a threshold measured from the global baseline with a stricter threshold computed from a local baseline. We adopted this method to reduce artefacts from the recording noise superimposed on the slow astrocytic dynamics and from slow changes in baseline due to physiological or imaging drifts. All the $Ca^{2+}$ traces were visually inspected to exclude the ROIs dominated by noise. For all experiments, we calculated the number of active ROIs and for each ROI corresponding to the soma, proximal processes and microdomains the frequency, and the amplitude of the $Ca^{2+}$ signal. All the $Ca^{2+}$ peaks were aligned to their onset to compute the average $Ca^{2+}$ peak (Supplementary Fig. 17). The onset of each $Ca^{2+}$ event was defined as the last time point when its fluorescence trace was below one standard deviation of the baseline. Finally, the time onset of all detected $Ca^{2+}$ events was reported in raster plots and peristimuls time histograms (PSTH). These procedures were applied for the analysis of both in vivo and brain slice data. To provide an estimate of the change in the overall microdomain activity per astrocyte following PV or SST interneuron stimulation, the number of individual micro-domains (active ROIs) and the average frequency of $Ca^{2+}$ microdomain events per cell were measured under the different experimental conditions. Then, these values were averaged across all astrocytes to obtain the bar graphs reported in the figures. A response ratio (RR; Fig. 8a) that describes the change in the response of astro-cytic processes to successive stimuli was calculated as follows. Firstly, for each astrocyte the response to a given interneuron stimulation was measured by the number of active ROIs, frequency and amplitude of $Ca^{2+}$ peaks at proximal and fine processes. These values were normalized to their corresponding baseline values, pooled and averaged. Secondly, the astrocyte RR was defined as the ratio of the second to the first response to 10 (or 30) pulse activation of PV or SST interneurons.

Mean IPSCs peak amplitudes in Supplementary Fig. 13 were fitted to the double exponential equation $A(t) = A1*\exp(-x/t1) + A2*\exp(-x/t2)$, where $A1$ and $A2$ are the amplitude of the fast and the slow decay component and $t1$ and $t2$ are the corresponding decay time constants.

**Statistical analysis.** Data were tested for normality before statistical analysis. For the number of ROIs and the frequency of astrocytic $Ca^{2+}$ events, we used paired Student's $t$-test (on normal data distribution) or paired sample Wilcoxon signed-rank test (on non-normal data distribution). For the RR, one sample Wilcoxon signed-rank test was used. For cumulative distribution comparisons, we applied the Kolmogorov–Smirnov test. For EM data, normality test and statistical analysis were performed using GraphPrism v.4.0 (GraphPad Software, San Diego, CA, USA). Given the non-normal distribution of data, Mann–Whitney test and Kruskal Wallis with Dunn's multiple comparison test were used. Pairwise statistical comparisons of each value of the astrocyte response to a given stimulation was carried out with respect to basal values. The astrocytes response to the two successive 10 (or 30) pulse stimulation was also similarly evaluated. All results are presented as mean ± SEM. Results were considered statistically significant at $p \leq 0.05$. *$p \leq 0.05$, **$p \leq 0.01$, ***$p \leq 0.001$. The exact $p$-values for each set of data are reported in the Supplementary Table 4.

**Data availability.** Data presented in this work are available from the corresponding author upon reasonable request.

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

## Acknowledgements

We are grateful to T. Pozzan, S. Vicini, and P. Magalhaes for their valuable comments on the manuscript and discussions. We also thank B.S. Khakh and R. Srinivasan for help in the use of the GECIquant software. The work was supported by Telethon Italy Grant GGP12265, Cariparo Foundation, National Research Council Aging Project, Fondo per gli Investimenti della Ricerca di Base Grant RBAP11×42L, PRIN 2015-W2N883_001 and Marie Skłodowska-Curie ITN, EU-GliaPhD to G.C., Istituto Italiano di Tecnologia, European Research Council (ERC, NEURO-PATTERNS), FP7-Health (DESIRE), and MIUR FIRB (RBAP11×42L) to T.F. and PRIN (2010JFYFY2), INRCA intramural funds, UNIVPM, and Fondazione di Medicina Molecolare to F.C.

## Author contributions

L.M. performed in vivo Ca$^{2+}$ imaging experiments with the collaboration of M.G.-G. and A.C., developed the software for Ca$^{2+}$ data analysis and analyzed data. G.L. and L.M. performed the Ca$^{2+}$ imaging experiments in brain slices and analyzed astrocytic Ca$^{2+}$ signals with the collaboration of M.Z. and A.L.. G.L., L.M., M.S., and I.M. performed the patch-clamp recording experiments in brain slices. M.S., I.M., A.L., S.B., and L.M.R. performed AAV injections and control experiments in brain slices. C.V. provided the SSR4 KO mice and B.B. the GABA$_{B2}$ KO mice. F.C. and T.F. contributed to the design of EM and electrophysiological experiments in vivo that were performed by M.M., with the contribution of A.P. and A.F., respectively. A.C. and S.B. performed control experiments on ChR2 and GCaMP6f expression. G.C. designed the study and wrote the manuscript with inputs from all the authors.
