## [Peer Review File · Nature Communications]

Reviewers' expertise:

Reviewer #1: GABAergic interneurons, in vivo imaging, optogenetics, electrophysiology;

Reviewer #2: Neuron-astrocyte interaction;

Reviewer #3: Neuron-astrocyte interaction, in vivo imaging;

Reviewer #4: Neuron-astrocyte interaction, in vivo imaging, optogenetics.

Reviewers' comments:

Reviewer #1 (Remarks to the Author):

The paper by Mariotti et al. examines how two types of GABAergic interneurons, parvalbumin+ and somatostatin+ interneurons, signal to astrocytes. To address this question, the authors use virus injection, electrophysiology, optogenetics, and two-photon imaging in vitro and in vivo. The main findings are:

- Stimulation of GABAergic interneurons induces GABAB receptor-mediated Ca²⁺ elevations in astrocytes.

- Ca²⁺ responses evoked by stimulation of PV+ interneurons are weak and depress, whereas responses upon stimulation of SOM+ interneurons are stronger and facilitate during repeated stimulation.

- The potentiation of Ca²⁺ responses evoked by SOM+ interneuron stimulation is dependent on release of the neuropeptide somatostatin.

Based on these results, the authors conclude that they have identified a cell type-specific signaling mechanism from GABAergic interneurons to glial cells. Although the paper contains certain interesting aspects, I regret to say that the manuscript is clearly not suitable for publication in a high-profile journal for two reasons. First, the physiological significance of differential signaling remains unclear. Second, the manuscript is preliminary, and several problems need to be addressed, as detailed below.

Major points:

1. The authors apparently think that the Ca²⁺ transients the astrocytes are generated by direct interactions between GABAergic synapses and astrocyte processes. However, they could be equally mediated by network effects, especially in the optogenetics experiments where multiple interneurons are stimulated. The long response latency, sometimes several seconds, would be consistent with this idea. Repeating experiments in the presence of blockers of excitation would be needed to distinguish between these possibilities.

2. One of the main points of the paper is that the astrocytic Ca²⁺ transient is potentiated after repetitive stimulation of SOM+ interneurons. However, this finding is not entirely convincing. First, the differences are small, and pairwise statistical comparisons are not carried out in all cases. Second, the stability of the Ca²⁺ transients is unclear. Long-lasting control experiments without stimulation should be provided. Finally, it is not clear how the amount of potentiation depends on the temporal order of short versus long stimuli.

3. The physiological significance of the observed effects is unclear. The authors argue that

the potentiation of the glial Ca²⁺ transients may represent a “memory mechanism”, but no evidence for this provocative idea is provided. First, GABAergic cells are often weakly tuned to external stimuli and thus do not encode information. Second, even if that were the case, it is totally unclear how that memory trace would be read out. Finally, it is not clear whether the potentiation is long-lasting. All these aspects must be clarified experimentally before the authors can talk about a memory mechanism.

4. The in vivo measurements suffer from the problem that they were recorded in the presence of an anesthetic / pharmacological cocktail (urethane, dexamethasone, and atropine). Ideally, experiments should be carried out in awake animals. At the very least, this serious limitation of the in vivo part of the paper should be clearly mentioned.

5. The authors stimulate interneurons with 10 or 30 light pulses, each of which has a duration of 150 ms and triggers several action potentials. Whether such an activity can occur under physiological conditions is not entirely clear, at least for the SOM+ interneurons.

6. The molecular mechanisms underlying the Ca²⁺ transients are unclear, and the link between GABAB or somatostatin receptors and Ca²⁺ signals also remains unidentified. A basic pharmacological characterization should be included. For example, the hypothesis of an involvement of TRP channel should be tested using selective TRPA1 blockers. It seems important to know whether the same mechanisms are involved in soma, proximal processes, and microdomains, and whether there are any differences depending on whether Ca²⁺ transients are triggered by PV+ and SOM+ interneurons.

7. Finally, the presentation of the data leaves room for improvement. First, data must be presented more precisely and quantitatively (examples given below). Second, the Methods section is excessively long because of several redundancies. It should be rewritten to present the facts more concisely.

Minor points:

Line 24 ff: The excessive use of “respectively” throughout the paper makes the manuscript unnecessarily difficult to read.

Line 62: “targeted” is misspelled.

Line 62 ff: These technical details that should go into the Methods section. What is “vxc” ?

Line 80: “few Ca²⁺ elevations” should be replaced by quantitative data.

Line 80 ff: The overused word “significant” should be reserved for statistically significant differences. It doesn’t sound this was the case.

Line 88 ff: The vague term “interneuron signaling” needs to be replaced.

Line 103: “GABAB ...” - At this point the receptors mediating the Ca²⁺ responses have not

been identified.

Line 106: What is the evidence that the Ca²⁺ transients are oscillatory?

Line 117: "Cumulative distribution of Ca²⁺ events" - I assume the authors talk about event frequency here, but this should be more explicitly stated.

Line 120: "memory mechanism" - as stated above, it is unclear whether this "memory" is physiologically relevant.

Line 131: "While this process needs to be further investigated in specifically designed experiments" - such kind of data should be included in the present manuscript.

Line 176: "absolute mean amplitude of IPSCs ... found to be lower" - as synapses by SOM+ interneurons are formed on distal dendrites, the observed differences may be accounted for by voltage-clamp errors and tell us little about GABA release.

Line 186: "PV interneurons release only GABA" - how do the authors know this? For example, PV+ interneurons have been shown to express corticotropin releasing hormone (CRH; Yan et al., 1998, Hippocampus), and their presynaptic terminals contain large dense core vesicles (Ramirez-Franco et al., 2016, Frontiers).

Line 203: What was the concentration of the antagonist? This should be mentioned in the text, not only in the legend.

Line 220: The pharmacology results show that somatostatin receptors are involved in the glial Ca²⁺ responses, but this does not necessarily mean that somatostatin activates them. The authors should draw their conclusions more carefully.

Line 322: "generates cell-specific interneuronal-astrocytic networks" - misleading, because the networks are constitutively present.

Line 345: I only know "xylazine".

Line 360: What was the exact temperature range for the "room temperature" recordings?

Line 374: "typically" should be replaced by exact numbers.

Line 421 ff: The text is partially redundant with previous statements and should be shortened.

Line 474: "According to with" - sentence screwed up.

Line 494: Rigorous immunocytochemistry requires that the specificity of the antibodies was tested in KO animals. Was this done for all antibodies?

Line 538: "the anti-GFP antibodies were used to label GCaMP6f" - inaccurate, revise.

Line 591: The figures are bar graphs, not histograms.

Line 605: "Kruskal Wallis" is misspelled.

Figure 2: A more systematic comparison between the first and the second stimulus should be performed.

Figure 2, legend: Do I understand correctly that the bar graphs pooled data from in vivo and in vitro measurements? Clearly, the two data sets were obtained under very different conditions and should be separated. Also, "Kolmogorov" is misspelled.

Figure 5, legend: "perisynaptic" is misspelled.

Figure 7, legend: "take the stage" sounds unscientific and should be replaced. Furthermore, the scheme shown in panel b is silly – it should be extended or deleted.

Throughout all legends, the authors should specify precisely whether the data shown were recorded in vitro or in vivo.

Supplementary figure 1: the colocalization doesn't say very much, because both tdTomato and eYFP are Cre-dependent. It would be more interesting to perform double labeling with PV and somatostatin antibodies.

Reviewer #2 (Remarks to the Author):

In the present work authors have investigated the astrocyte calcium responses to parvalbumin (PV) and somatostatin (SST) interneurons. They have used state-of-the-art techniques, including two photon microscopy, cell specific virus expression, electrophysiology and optogenetics in slices and in vivo.

This is an important, clear and well-performed study which adds valuable information regarding astrocyte-neuron interaction, a relevant and emerging, yet debated, topic in neuroscience.

Authors show that astrocytic responses depress upon Parvalbumin interneuron repetitive stimulations and potentiate upon Somatostatin interneuron repetitive stimulations. They also show that this latter potentiation was mediated by activation of astrocytic somatostatin receptors. Authors conclude that astrocytes display plasticity in response to interneuron activation.

The results are novel and interesting. However, some conclusions stated in the text are not

sufficiently supported by current experimental data provided. Specifically, the major conclusion of the manuscript, i.e., that astrocytes display plasticity responses, requires further analysis. As detailed below, although this is an interesting and likely possibility, further analysis providing stronger evidence are required to exclude the possibility that this is simply due to plasticity changes in synaptic transmitter release.

Therefore, I feel that present manuscript has potential merit to be published, but some important concerns need to be addressed.

Major comments

The major concern stem from the conclusion of astrocyte response plasticity because it is insufficient data to exclude the possibility that the different astrocytic responses were due to changes in synaptic transmission. A more detailed analysis of synaptic properties over time needs to be provided by monitoring IPSC amplitude to check for potential synaptic changes. Specific comments on this issue:

1. Page 4, line 85, it is stated "a second 30 pulse stimulation evoked reduced Ca²⁺ elevations indicating a depression of the astrocyte response to successive episodes of PV interneuron activity". Alternatively to astrocytic plasticity, this phenomenon may be due to 1) reduced IN response to light or 2) synaptic depression of GABA release from the interneuron. Authors should provide a figure displaying the quantification of 1) action potential firing and 2) IPSC amplitude over time during the stimuli.

Authors seem to have discarded potential synaptic changes based on data presented in Supplementary Fig. 4. However, the results shown are unclear and insufficient to support the conclusion, as detailed in point 7 below.

2. Page 4, line 100, it is stated "These data indicate that astrocytes are more sensitive to SST than PV interneuron signaling". Again, rather than different sensitivity, this may alternatively explained by either different release probability properties and different amount of activated nerve terminals. Again, authors should provide the analysis of IPSC amplitudes in pyramidal neurons over time during the stimuli, to test for potential IPSC amplitude changes reflecting synaptic changes.

3. Page 4, line 103. Authors conclude that the longer delays observed indicate slow intracellular GABABR-mediated signalling pathways. This may be indeed the case as concluded by the authors. But, alternatively, these delays may be due to the fact that large GABA concentrations are required to be built up to reach GABAB receptors in astrocytes. Does the duration of the stimulus influence the delay of the response? E.g., if shorter pulses have the same delay, author's interpretation would be correct; otherwise, the alternative interpretation would seem more adequate. Whatever the mechanistic interpretation, it does not compromise author's results, but they either test alternative interpretations or tone down the conclusion presenting the alternative interpretations.

4. Page 5, line 105, "they sense tonic elevations of ambient GABA". This again might be true, but no data is presented in the manuscript supporting this idea. That statement would

bee more appropriate in the discussion section.

5. Page 5 line 114, similar to the concern in point 1, the conclusion that there is “a potentiation of the astrocyte response to SST interneuron signaling” needs to be confirmed by excluding potential changes in firing rate and synaptic transmission over time during optogenetic stimulation. Figures displaying firing rates and IPSC amplitude over time need to be shown.

6. Page 5 line 118, authors conclude: “Altogether, these data demonstrate that astrocytes modulate

their Ca²⁺ dynamics coherently with the previous history of interneuron network activity unveiling in these non-neuronal cells a memory mechanism of the past events which occurred in the GABAergic interneuron networks.” This a very important conclusion that therefore requires the proper support by excluding changes in GABAergic transmission, as specifically indicated in points 1 and 5.

7. Supplementary Fig 4 is unclear; e.g., which IPSC is measured, the first or the last IPSC in the train? The important data is actually not shown, and plots of the individual IPSC amplitude over time (for both the first and second train of stimuli) need to be shown.

Changes due to short term facilitation or depression need to be checked.

Also, while the authors measured IPSC charge, the analysis of the IPSC peak amplitude seems more appropriate because this parameter would more directly reflect the amount of GABA released (the charge may be affected by the activity of GABA transporters).

In summary, authors need to show both actual and averaged firing rate as well as IPSC amplitudes for PV and SST stimulation for the first and second train over time, to assess whether changes in synaptic transmitter released occurred that may alternatively explain the observed changes in astrocytic responsiveness.

Minor comments

1. There are several acronyms and terminology used throughout the text that is not defined or unclear. For example, Page 3, line 65, what is vxc? On Page 7, line 159, define G42.

2. When referencing GABAergic signaling into astrocytes in page 6, line 149, additional original references on this issue should be included: Navarrete and Araque 2008; Gould et al. Philos Trans R Soc Lond B 2014; Perea et al. eLife 2016.

3. Page 10, line 252, when discussing Ca²⁺ signal regulation evoked by neuronal signals, the references Perea and Araque, J Neurosci 2005 and Schipke et al, Cer Cortex 2008 may be included.

Reviewer #3 (Remarks to the Author):

The manuscript by Mariotti and colleagues describes a novel pathway of neuron-glia

communication.

While the role of the major type of glia, the astrocytes, in transmitter homeostasis via respective uptake systems is well established, we are just at the beginning to understand why astrocytes simultaneously express receptors to the same transmitter, often with different intracellular signaling pathways as adjacent neurons. Recent studies provided strong evidence for a fast responsiveness of hippocampal astrocytes to the excitatory transmitter glutamate.

Here, Mariotti et al. now focused on the role of inhibitory, GABAergic interneurons (INs) in the somatosensory cortex (SSC) and their communication to nearby astrocytes. They took advantage of genetically modified mice with cell-specific fluorescent protein or Cre DNA recombinase and AAV-mediated induction of channelrhodopsin or calcium indicator expression to study interneuron-astrocyte signaling in vivo with minimal perturbation. Interneurons form a diversity of subclasses that often are intermingled in the same region of the brain, e.g. in the same cortical layer. Therefore, the authors used mice in which two type of interneurons were genetically modified, the parvalbumin (PV) and the somatostatin (SST) INs. The in vivo results were confirmed and complemented by experiments in acutely isolated brain slices of the same brain region.

This study provides important confirmatory and novel results.

1. The authors further confirm (here, for the SSC), the expression of GABAB receptor on astrocytes and its link to intracellular Ca^{2+} changes, which is in contrast to the neuronal signaling.
2. Different to the astroglial response to glutamate, GABA released from stimulated INs does not elicit immediate Ca^{2+} responses. Bursts of activity are required to increase the frequency of local signals in the fine network of peripheral astroglial processes.
3. The most important observation is the distinct response of astrocytes to defined activation of different IN subpopulations. While PV IN can elicit only rather weak responses, SST neurons can synergistically boost their signal impact by co-releasing somatostatin. The authors can further show that astrocytes express the somatostatin receptor SSRT4 for this pathway. Distinct astroglial responses to defined subpopulations of interneurons is a frequently discussed topic, but never convincingly shown.
4. The data further provide evidence that accumulation of small Ca^{2+} signals in peripheral processes is integrated by the cell soma.

The manuscript is well written. The methodology is state-of-the art and has been carefully performed. The results are very carefully discussed.

So far, this study is still a description of a carefully performed experiments. Additional results will be required to identify cortical microcircuits in which IN-evoked increases of Ca^{2+} signals play functional roles. However, such experiments go far beyond the current study. However, the authors should consider the discussion of it.

Since the Ca^{2+} signals are rather small, the illustration is rather difficult. The authors should try to apply a color LUT, rather than a grey scale LUT to their movies.

Reviewer #4 (Remarks to the Author):

Mariotti et al. investigated the existence of selective interneuron-astrocyte signaling in adult neocortex by combining optogenetics with 2-photon Ca imaging in in vivo and in situ slice studies. They focused the study on 2 interneurons subtypes, Parvalbumin (PV) and Somatostatin (Som) expressing interneurons, and they found that these interneurons differentially signal to astrocytes. Thus, Som induced robust GABAB receptor-mediated Ca elevations, while PV induced weak Ca signals. The analysis of the responses showed that enhanced responses driven by Som were mediated by the combined release of somatostatin peptide and GABA. In summary, this work shows very interesting data suggesting cell-specific interneuronal-astrocytic networks.

There are still some major concerns:

- 1) There is a frequent absence of direct evidence of GABAergic actions onto astrocytes. All the results rely on the GABA released from interneurons activates GABA_B receptors in astrocytes. However, because the experiments were done in absence of blockage of other neurotransmitter receptors (i.e., glutamate), the particular connectivity of PV- SOM would result in a disinhibition of the local inhibitory network, which might cause the boosting calcium responses of astrocytes derived from glutamatergic signaling. Thus, indirect effect of GABA through neuronal network would underlie the enhanced Ca signals in astrocytes after 10 and 30 s stimulation protocols.
- 2) Authors show IPSC charge after light stimulation; however, it's unclear how they measured it. Considering that changes in astrocyte Ca signals happen after 10-20s after interneuron stimulation, it would be necessary to test the time course of GABA levels (IPSCs) to ensure no changes in the inhibitory synaptic transmission.
- 3) There are some confusing data regarding the recording and analysis of cell soma Ca signals. In vivo experiments shown in Figure 1 display astrocyte Ca increases at the soma after PV and Som interneuron stimulation, but no statistically significant, Figure 2. "Based on the sparse nature of GCaMP6f expression in astrocytes", authors change the approach and use Fluo-4 loaded astrocytes to record somatic Ca signals, and then they found a robust effect for both PV and Som interneurons, Figure 3, 4. However, they use again the GCaMP6f expression to monitor the effects of somatostatin peptide on the Ca responses in microdomains, proximal processes and cellular soma, Figure 6. In that case, the found robust responses in all of them. Therefore, what is the final message? Do PV and Som interneurons induce Ca signals at the astrocytic soma? The information shown in Figure 2, which is the average of in vivo and slice data, has to be revised as well as the message of the text.
- 4) As authors indicated in the abstract interneurons "differentially signal to astrocytes at fine processes weak and robust GABAB receptor-mediated Ca elevations"; however, they did not test the dependence of local Ca signals to GABA_B receptor activation. They analyzed

somatic Ca signal induced by intense protocols of stimulation of interneurons (30); but, were microdomains sensitive to SCH? Considering the importance of microdomains as key points in close contact with synapses, it would be interesting to elucidate whether GABA_B antagonist can abolish the observed responses by moderate (10) and stronger (30) stimuli.

5) Figure 6a: the actual data cannot exclude the effects of somatostatin peptide on neuronal network. To ensure that somatostatin peptide induces Ca signals in astrocytes by activation of SSTR4 receptors a cocktail of antagonists for different neurotransmitter receptors (plus TTX) should be used.

6) The modulatory role of somatostatin peptide in the astrocyte Ca signal is an interesting result. If so, would somatostatin peptide boost the Ca responses to local application of GABA? Could PV stimulation plus somatostatin peptide (in the bath) mimic the astrocyte responses to Som interneurons?

7) Although the idea is tempting, there is no evidence from the actual data to support the author's conclusion of the existence of a cellular memory mechanism in astrocytes. -Proper controls of neuronal network activity during and after PV and Som stim are required to discard that the increase of spontaneous Ca oscillations in astrocytes is derived from network activity.

-What are the mechanisms regulating this memory?

-Do astrocytes modulate their GABA_B receptor or GABA transporter levels in the membrane as response to the history of GABAergic network activity?

Minor points:

Figure 6a mentions Ca signals induced by somatostatin peptide, but not by somatostatin interneurons.

The analysis of response ratio (RR) shown in Figure 7 is confusing. Why authors put together signals coming from microdomains and proximal processes that showed different behavior to interneuron stimulation?

From the methods is unclear to me whether authors summarized values of frequency and amplitude from those structures in a single data point. If so, why?

Reviewer #1 (Remarks to the Author):

We thank the reviewer for his/her constructive comments and suggestions that have significantly improved the manuscript.

Major points:

1. *The authors apparently think that the Ca²⁺ transients the astrocytes are generated by direct interactions between GABAergic synapses and astrocyte processes. However, they could be equally mediated by network effects, especially in the optogenetics experiments where multiple interneurons are stimulated. The long response latency, sometimes several seconds, would be consistent with this idea. Repeating experiments in the presence of blockers of excitation would be needed to distinguish between these possibilities.*

The point raised by the reviewer is important. SST interneurons are known to inhibit PV interneuron firing and, in turn, to decrease PV interneuron-mediated inhibition of pyramidal neurons ultimately increasing glutamatergic excitation in local circuits. Although this disinhibitory mechanism is probably less effective in the Somatosensory cortex layer 2/3 with respect to other cortical layers {Xu, 2013 #10916} and brain regions {Pfeffer, 2013 #10965}, we agree with the reviewer that it has the potential to contribute to enhance astrocytic Ca²⁺ elevations in response to optogenetic SST interneuron signalling. Results from the experiments that we performed following the reviewer's suggestion revealed that in the presence of blockers of glutamatergic excitatory signals, such as NBQX and AP5, i.e., specific antagonists of the ionotropic glutamate AMPA and NMDA receptors, respectively, and MPEP, i.e., a specific antagonist of type 5 metabotropic glutamate receptors (these latter representing a main pathway for the astrocyte Ca²⁺ response to synaptic glutamate), the overall Ca²⁺ response of GCaMP6f-expressing astrocytes to SST interneurons was unchanged. These data are reported in the new Suppl. Fig. 10 of the revised manuscript.

Please note that a set of new experiments that we performed provide evidence that the increase in Ca²⁺ microdomains activity induced by PV or SST interneuron activation was blocked after slice perfusion with the specific GABA_BR antagonist SCH50911 (see Suppl. Fig. 7). These latter results complement those reported in the original manuscript (see original Fig. 3c-e and Fig. 4a-c) on the inhibition by SCH50911 of the somatic Ca²⁺ response.

All together, these results provide further support to the conclusion that the Ca²⁺ transients evoked in astrocytes by GABAergic interneuron stimulation are unlikely due to network effects and they rather depend on a direct activation of astrocytic GABA_B receptors.

2. *One of the main points of the paper is that the astrocytic Ca²⁺ transient is potentiated after repetitive stimulation of SOM+ interneurons. However, this finding is not entirely convincing. First, the differences are small, and pairwise statistical comparisons are not carried out in all cases. Second, the stability of the Ca²⁺ transients is unclear. Long-lasting control experiments without stimulation should be provided. Finally, it is not clear how the amount of potentiation depends on the temporal order of short versus long stimuli.*

i). Pairwise statistical comparisons of each value of the astrocyte response to a given stimulation was carried out with respect to basal values. The astrocytes response to the two successive 10 (or 30) pulse stimulation was also similarly evaluated. Please note that in the new Fig. 2 summarizing the results from in vivo experiments (and the new Suppl. Fig. 7 summarizing the results from slice experiments), statistically significant values were marked by asterisks, while the absence of asterisks indicates that, according to the pairwise comparison, the differences in the values were not statistically significant.

ii). We agree with the reviewer that given the recognized light-sensitivity of astrocytes, to avoid misinterpretation of the results obtained from single and two-photon laser-scanning microscope Ca²⁺ imaging experiments, appropriate control experiments are necessary. We are well aware of this and in our experiments we always used a low laser power at the sample (between 5 and 10 mW, as we reported on line 389 of the original manuscript). Please note that in our original study, we already performed the control experiments that the reviewer requested in order to check whether the sequences of astrocytic Ca²⁺ signal imaging - without the optogenetic stimulation - perturbed *per se* astrocytic Ca²⁺ signals. We underevaluated the importance of mentioning the results from these control experiments in the original manuscript.

We have prepared a figure reporting the stability over time of the Ca²⁺ signal at the different cell compartments of GCaMP6f-expressing astrocytes *in vivo* and in slice preparations during imaging of astrocytes in the absence of optogenetic stimulation. Please note that we applied the same imaging protocol that was used for studying the astrocytes response to interneuron stimulation. Results from these experiments are reported in the new Supplementary Fig. 9a,b and commented on line 128-132.

iii) To answer the final question raised by the reviewer in point 1, we performed a new set of experiments (7 GCaMP6f astrocytes, 6 slices, 3 mice) and results obtained suggest that the amount of potentiation of the astrocyte response to the successive 30 light pulse stimulations of SST interneurons, with or without previous 10 light pulse stimulations of the same interneurons, is similar (1.40 ± 0.10 and 1.47 ± 0.41 , respectively). Although these data should be interpreted with caution because of the limited number of cells studied, they suggest that the amount of potentiation to the successive 30 light pulse stimulations of SST interneurons is independent on whether or not the 30 light pulse stimulation was preceded by shorter periods of activity in the same interneurons.

We may have, however, misunderstood the comment of the reviewer. Perhaps the reviewer was referring to the possibility that the response of astrocytes to a first episodes of 30 light pulse stimulation can be reduced if this stimulation is not preceded by two episodes of 10 light pulse stimulation. Figure 12 (see also Fig. 3) reports the results from experiments in which 30 light pulse stimulations of SST interneurons were performed without previous 10 light pulse stimulations. These results show a highly significant increase in somatic Ca²⁺ elevations in astrocytes in response to SST interneurons.

3. The physiological significance of the observed effects is unclear. The authors argue that the potentiation of the glial Ca²⁺ transients may represent a “memory mechanism”, but no evidence for this provocative idea is provided. First, GABAergic cells are often weakly tuned to external stimuli and thus do not encode information. Second, even if that were the case, it is totally unclear how that memory trace would be read out. Finally, it is not clear whether the potentiation is long-lasting. All these aspects must be clarified experimentally before the authors can talk about a memory mechanism.

We cannot but agree with the reviewer entirely that it will be important to understand the possible physiological significance and identify the molecular mechanism of the presumed cellular memory form in astrocytes. We believe, however, that to address the points raised by the reviewer, a complex *in vivo* approach, preferentially in non-anesthetized mice, is necessary. At present, we are not fully equipped to appropriately develop this approach but this certainly represents a main goal of our future research. Please note that we toned down our comments about this issue and the term “*memory*” is mentioned in the text only once at page 12, line 292, as follows:

"... astrocytes in the adult mouse somatosensory cortex modulate their Ca²⁺ responses as a function of previous states of activity in the surrounding interneuronal network suggesting the existence of a form of cellular memory in astrocytes."

4. The *in vivo* measurements suffer from the problem that they were recorded in the presence of an anesthetic / pharmacological cocktail (urethane, dexamethasone, and atropine). Ideally, experiments should be carried out in awake animals. At the very least, this serious limitation of the *in vivo* part of the paper should be clearly mentioned.

Following the comment of the reviewer, on page 13, line 320-322 of the Discussion, we state that:

"It will be also important to validate the GABAergic interneuron type-specific response of astrocytes in awake non-anaesthetized animals given that astrocyte Ca²⁺ signalling can be affected by anaesthetics {Nimmerjahn, 2009 #8627}{Thrane, 2012 #10551}."

5. The authors stimulate interneurons with 10 or 30 light pulses, each of which has a duration of 150 ms and triggers several action potentials. Whether such an activity can occur under physiological conditions is not entirely clear, at least for the SOM+ interneurons.

The reviewer's concern is reasonable. We cordially invite the reviewer to consider the arguments below that support the similarities between the firing frequency rates of SST interneurons reported in our experiments and those reported in three representative studies in awake mice published in high-profile journals.

In our study we evaluated the firing discharge of SST interneurons following either optogenetic light pulses (to groups of cells) or depolarizing current pulses (to individual cells). During the overall period of stimulation by light pulses *in vivo* and current pulses in slices, the mean frequency rate of Somatostatin interneurons was 13.4 ± 2.41 Hz (page 4, line 87 and Suppl. Fig. 3) and 9.0 ± 1.44 Hz (see legend of Fig. 4), respectively. Please note that, notwithstanding this difference in the firing rate, the two stimulation protocols evoked in astrocytes comparable Ca²⁺ responses.

In a representative study by the group of Carl Petersen (*Gentet et al. Nature Neuroscience 2012, 15:207-212*){Gentet, 2012 #10912}, the mean frequency rate of SST interneurons in awake head-restrained mice during quiet wakefulness was 6.3 ± 0.6 Hz (range, 1 to 16 Hz).

In Adesnik et al. (*Nature 2012, 490, 226-231*){Adesnik, 2012 #10487}, the mean firing of SST interneurons in awake running mice was 26.2 ± 2 Hz. Notably, this mean value was reduced tenfold (2.7 ± 0.4 Hz) in anesthetized mice.

In Pollack et al. (*Nat Neuroscience 2013, 16, 1331-1339*), SST interneurons increase during locomotion both spontaneous (11.6 ± 10.8 Hz) and visually evoked (28.6 ± 23.6 Hz) mean firing rates (see Suppl. Tables 1 and 2).

The firing rates that we induced by optogenetic light pulses to SST interneurons are, therefore, comparable to those observed in these interneurons from awake mice. These studies are now cited on page 4, line 88.

As regards the peak firing rates of SST interneurons, we obtained values of 73.1 ± 13.1 Hz during the 150 ms light pulses (Suppl. Fig. 3) and 30 ± 4.8 Hz during the 300 ms current pulses to individual interneurons (Fig. 4 legend) and these values are also comparable to those obtained from a new set of experiments in which by juxtosomal recordings we evaluated the firing discharge of SST interneurons from layer 2 somatosensory cortex of awake mice. Indeed, during quiet wakefulness (monitored by a camera combined with infrared illumination), SST interneurons exhibit a bursting discharge behavior with firing frequency (mean, 42.7 ± 4.4 Hz, range 29-57; Suppl. Fig. 5) that is higher than that observed in our experiments following current pulses to SST interneurons (30 ± 4.8 Hz). Also noteworthy is that SST interneurons exhibit at rest a bursting behavior during 15 % of their

activity that correspond to the timing of the bursts induced by the optogenetic light pulse to SST interneurons in our experiments (150 ms pulse duration at 1 Hz). For reviewer inspection, we include below a Table that reports a summary of *in vivo* awake recordings of spontaneous firing activity of 7 somatostatin expressing cells in somatosensory cortex.

Cell	Age	Depth (μm)	Rec time (s)	Ave Freq. (Hz)	Peak isi (ms)	Freq (Hz)	Burst duration
20170502c	53	180	304.6	4.4	17.5	57.1	28.3%
20170502d	53	170	172.4	5.2	27.5	36.4	17.3%
20170503a	54	140	96.3	3.9	17.5	57.1	19.4%
20170503b	54	150	202.8	6.3	22.5	44.4	18.8%
20170504a	55	180	256.6	0.8	35	28.6	4.2%
20170504b	55	160	540.0	1.8	32.5	30.8	5.0%
20170504c	55	170	300.0	0.6	22.5	44.4	14.8%
Average	54.1	164.3	267.5	3.3	25.0	42.7	15%
SEM	0.3	5.7	53.3	0.8	2.6	4.4	3%

Altogether, these observations suggest that the action potential discharge induced to Somatostatin interneurons in our study, in terms of the bursting behavior and mean firing rate, is consistent with the activity of these interneurons in awake behaving mice.

6. The molecular mechanisms underlying the Ca^{2+} transients are unclear, and the link between GABAB or somatostatin receptors and Ca^{2+} signals also remains unidentified. A basic pharmacological characterization should be included. For example, the hypothesis of an involvement of TRP channel should be tested using selective TRPA1 blockers. It seems important to know whether the same mechanisms are involved in soma, proximal processes, and microdomains, and whether there any differences depending on whether Ca^{2+} transients are triggered by PV+ and SOM+ interneurons.

Following the reviewer's comment, we addressed the hypothesis that TRP channels contribute to astrocytic GABA_BR-mediated Ca^{2+} transients. We used HC-030031, a specific antagonist of TRPA1 channels that previous studies suggested to be expressed in astrocytes and contribute to control Ca^{2+} microdomain activity (Shigetomi et al., Nat. Neuroscience 2012, 15, 70–80; Shigetomi et al., The Journal of Neurosci. 2013, 33, 10143-53). We found that the astrocyte Ca^{2+} response at soma, proximal processes and microdomains evoked by PV or SST interneuron stimulation was not significantly affected in the presence of HC-030031. These results are reported in the new Suppl. Fig. 8.

7. Finally, the presentation of the data leaves room for improvement. First, data must be presented more precisely and quantitatively (examples given below). Second, the Methods section is excessively long because of several redundancies. It should be rewritten to present the facts more concisely.

We modified the text according to the observations reported by the reviewer (see our point-by-point response below). According to the reviewer's request, we also shortened the Methods section.

Minor points:

Line 24 ff: The excessive use of “respectively” throughout the paper makes the manuscript unnecessarily difficult to read.

We reduced the use of "respectively", as requested.

Line 62: "targeted" is misspelled.
The typo was corrected.

Line 62 ff: These technical details that should go into the Methods section. What is "vxc" ?
We moved the sentence to the Methods. We removed "vcx" that was erroneously inserted.

Line 80: "few Ca²⁺ elevations" should be replaced by quantitative data.
We changed the sentence as follows: "...a more prolonged activation by 30 pulses induced a significant increase of both Ca²⁺ elevations at proximal processes and Ca²⁺ microdomain responses (Fig. 1b,c; see also Suppl. Fig. 6)." Please note that quantitative data of somatic Ca²⁺ elevations are reported in Fig. 2a and Suppl. Fig 7.

Line 80 ff: The overused word "significant" should be reserved for statistically significant differences. It doesn't sound this was the case.
We now use *significant* only to indicate a statistically significant difference.

Line 88 ff: The vague term "interneuron signaling" needs to be replaced.
We changed the subheading title into: *Successive episodes of SST interneuron activity evoke in astrocytes robust and potentiating Ca²⁺ responses.*

Line 103: "GABAB ..." - At this point the receptors mediating the Ca²⁺ responses have not been identified.
We modified the text accordingly.

Line 106: What is the evidence that the Ca²⁺ transients are oscillatory?
The Ca²⁺ response of astrocytes is now more appropriately defined as "*multiple Ca²⁺ elevations*".

Line 117: "Cumulative distribution of Ca²⁺ events" - I assume the authors talk about event frequency here, but this should be more explicitly stated.
We now specify on page 5 line 109, that rightward shift refers to "*...the cumulative distribution of Ca²⁺ event frequency*".

Line 120: "memory mechanism" - as stated above, it is unclear whether this "memory" is physiologically relevant.
We toned down the memory issue throughout the text and we modified the sentence on page 7, lines 156-159, as follows:
"*Altogether, these data demonstrate that astrocytes differently respond with GABA_BR-mediated Ca²⁺ elevations to PV and SST interneurons and change their response as a function of the previous history of activity in the surrounding GABAergic interneuron type-specific network.*"

Line 131: "While this process needs to be further investigated in specifically designed experiments" - such kind of data should be included in the present manuscript.
We agree with the reviewer that including in this manuscript further data on this important issue would be important. We believe that this can be a specific goal of a future project. Nevertheless, we have performed new *in vivo* experiments in which we evaluated somatic Ca²⁺ signals in a large number of astrocytes. The results obtained provide further support to

the existence of integrative properties of the astrocyte response to neuronal signals (see Fig. 3).

Line 176: “absolute mean amplitude of IPSCs ... found to be lower” - as synapses by SOM+ interneurons are formed on distal dendrites, the observed differences may be accounted for by voltage-clamp errors and tell us little about GABA release.

We agree with the reviewer that the poor voltage control of distal dendritic sites by somatic recordings can account for the difference between the IPSCs generated by SST and PV interneurons. Therefore, we removed the sentence regarding this point.

Line 186: “PV interneurons release only GABA” - how do the authors know this? For example, PV+ interneurons have been shown to express corticotropin releasing hormone (CRH; Yan et al., 1998, Hippocampus), and their presynaptic terminals contain large dense core vesicles (Ramirez-Franco et al., 2016, Frontiers).

Following the reviewer's criticism, we removed “... whereas PV interneurons release only GABA...” and modified the sentence as follows: “Given that SST interneurons release, in addition to GABA, the neuropeptide SST, we advanced the hypothesis...”

Line 203: What was the concentration of the antagonist? This should be mentioned in the text, not only in the legend.

The concentration of the antagonist is now reported in the text.

Line 220: The pharmacology results show that somatostatin receptors are involved in the glial Ca²⁺ responses, but this does not necessarily mean that somatostatin activates them. The authors should draw their conclusions more carefully.

We agree with the reviewer's comment. Accordingly, we draw more carefully our conclusions in general, and on the role of the neuropeptide Somatostatin, in particular.

Line 322: “generates cell-specific interneuronal-astrocytic networks” - misleading, because the networks are constitutively present.

Agreed. We modified the sentence as follows:

“...we identified a novel signalling mode of GABAergic interneurons that revealed the presence of cell-specific interneuronal-astrocytic networks.”

Line 345: I only know “xylazine”.

We corrected the typo.

Line 360: What was the exact temperature range for the “room temperature” recordings?

We have added the range of temperature.

Line 374: “typically” should be replaced by exact numbers.

We now specify that the access resistance was between 8.4 and 24.2 MΩ.

Line 421 ff: The text is partially redundant with previous statements and should be shortened.

We shortened this part of the text.

Line 474: “According to with” - sentence screwed up.

We removed “with”.

Line 494: Rigorous immunocytochemistry requires that the specificity of the antibodies was tested in KO animals. Was this done for all antibodies?

According to the datasheets provided by the suppliers, the antibodies were tested for specificity and displayed no detectable cross-reactivity with any other proteins and peptides used in our experiments. In particular, the anti-parvalbumin antibody (antibody registry ID on Resource Identification Portal:AB_477329, 1:1000 mouse, Sigma P3088) was derived from the PARV-19 hybridoma (raised against frog muscle parvalbumin) and recognizes a single 12-kDa protein on Western blot (Sigma Aldrich datasheet). The anti somatostatin antibody (antibody registry ID on Resource Identification Portal:AB_2255365, 1:200 rat, Millipore MAB 354) was raised against the synthetic 1-14 cyclic somatostatin and shows no cross-reactivity to enkephalins, other endorphins, substance P or CGRP. The specificity of anti parvalbumin anti somatostatin antibodies has been determined by preadsorption with the appropriate purified proteins and thoroughly described by others (Hackney et al., 2005, Xu et al., 2006).

Line 538: "the anti-GFP antibodies were used to label GCaMP6f" - inaccurate, revise.

We modified the sentence as follows: "*the anti-GFP antibodies were used to enhance the GCaMP fluorescence*".

Line 591: The figures are bar graphs, not histograms.

We substituted "histograms" with "bar graphs"

Line 605: "Kruskal Wallis" is misspelled.

We corrected the typo.

Figure 2: A more systematic comparison between the first and the second stimulus should be performed.

As we already mentioned in our response to point 2, pairwise statistical comparisons were regularly applied to evaluate the change in the astrocyte response between the two successive 10 (or 30) light pulse stimulations of interneurons. Please note that statistically significant values were marked by asterisks, while the absence of asterisks (instead of the addition of ns) indicates that the differences in the values were not statistically significant.

Figure 2, legend: Do I understand correctly that the bar graphs pooled data from in vivo and in vitro measurements? Clearly, the two data sets were obtained under very different conditions and should be separated. Also, "Kolmogorov" is misspelled.

The new Fig. 2 reports only the in *in vivo* data and the Suppl. Fig 8 reports only the brain slice data, as requested. Please note that these figures include data from additional experiments from both in vivo and slice preparations.

The type "Kolmorov" was corrected.

Figure 5, legend: "perisynaptic" is misspelled.

We corrected the typo.

Figure 7, legend: "take the stage" sounds unscientific and should be replaced. Furthermore, the scheme shown in panel b is silly – it should be **extended** or deleted.

We modified the title of Fig. 7 as follows: "Specificity of neuropeptide-releasing interneuron signalling to astrocytes." and extended the scheme reported in panel b, as suggested. We will remove this panel in the case the reviewer still considers it unsatisfactory.

Throughout all legends, the authors should specify precisely whether the data shown were recorded in vitro or in vivo.

We now add this information to each Figure legend.

Supplementary figure 1: the colocalization doesn't say very much, because both tdTomato and eYFP are Cre-dependent. It would be more interesting to perform double labeling with PV and somatostatin antibodies.

We performed new immunohistochemical experiments using antibodies for PV or SST interneurons, according to reviewer' suggestion. Results obtained revealed that 96.1 ± 2.10 (N = 3 animals) and 80.6 ± 2.36 % (N = 3 animals) of SST- and PV-positive interneurons, respectively, are also ChR2- positive. These data are reported in the modified Suppl. Fig 1.

Reviewer #2 (Remarks to the Author):

We thank the reviewer for the comments that "*This is an important, clear and well-performed study which adds valuable information regarding astrocyte-neuron interaction...*" and that "*The results are novel and interesting*". We also thank the reviewer for his/her constructive comments and suggestions that have significantly improved the manuscript.

The major concern of the reviewer regards the possibility that the reduction and the enhancement of the astrocyte response to PV and SST interneuron activity, respectively, can be due to a change in GABAergic transmission rather than to astrocyte response plasticity. We agree with the reviewer that to exclude this possibility "*A more detailed analysis of synaptic properties over time needs to be provided*". In our original study we performed a series of control experiments that addressed this possibility, and the results obtained were reported in Supplementary Fig. 4. The reviewer, however, stated that this figure is "*unclear and insufficient to support the conclusion*".

We regretfully admit that the results reported in Supplementary Fig. 4 could be confusing. According to reviewer' suggestions, in the new Suppl. Fig. 11 we now report the quantification of the action potential firing rate evoked in PV and SST interneurons by each light pulse of the two sequences of 10 and of the two sequences of 30 pulse stimulation as well as the peak amplitude (instead of the integral) of the evoked IPSCs in pyramidal neurons.

Specific comments of the reviewer on this issue:

1. Page 4, line 85, it is stated "*a second 30 pulse stimulation evoked reduced Ca²⁺ elevations indicating a depression of the astrocyte response to successive episodes of PV interneuron activity*". Alternatively to astrocytic plasticity, this phenomenon may be due to 1) reduced IN response to light or 2) synaptic depression of GABA release from the interneuron. Authors should provide a figure displaying the quantification of 1) action potential firing and 2) IPSC amplitude over time during the stimuli.

2. Page 4, line 100, it is stated "*These data indicate that astrocytes are more sensitive to SST than PV interneuron signaling*". Again, rather than different sensitivity, this may alternatively explained by either different release probability properties and different amount of activated nerve terminals. Again, authors should provide the analysis of IPSC amplitudes in pyramidal neurons over time during the stimuli, to test for potential IPSC amplitude changes reflecting synaptic changes.

The results of these important control experiments are now, at least in our opinion, more clearly reported in the new Suppl. Fig. 11 and they rule out the possibility that the different astrocyte response to PV and SST interneurons could be due to a change in the firing discharge of the two interneuron classes upon the successive optogenetic light pulse stimulations or to a change in synaptic GABA release.

3. Page 4, line 103. Authors conclude that the longer delays observed indicate slow intracellular GABABR-mediated signalling pathways. This may be indeed the case as concluded by the authors. But, alternatively, these delays may be due to the fact that large GABA concentrations are required to be built up to reach GABAB receptors in astrocytes. Does the duration of the stimulus influence the delay of the response? E.g., if shorter pulses have the same delay, author's interpretation would be correct; otherwise, the alternative interpretation would seem more adequate. Whatever the mechanistic interpretation, it does not compromise author's results, but they either test alternative interpretations or tone down the conclusion presenting the alternative interpretations.

According to the suggestion of the reviewer, we now report on lines 119-124 the delay of the astrocyte Ca²⁺ response to 10 and 30 light pulse stimulation as follow: *"The microdomain responses were detected with a delay of 14.93 ± 1.34 s from the onset of 10 pulse activation of SST interneurons and with longer delays at proximal processes and soma which may reflect slow intracellular GABABR-mediated signalling pathways. A similar delay (14.97 ± 1.30 s) was measured following the onset of 30 pulse activation of SST interneurons (Supplementary Fig. 4) suggesting that the duration of interneuron stimulation does not affect the delay of the astrocyte response."*

4. Page 5, line 105, "they sense tonic elevations of ambient GABA". This again might be true, but no data is presented in the manuscript supporting this idea. That statement would be more appropriate in the discussion section.

We moved this statement to the discussion, as suggested.

5. Page 5 line 114, similar to the concern in point 1, the conclusion that there is "a potentiation of the astrocyte response to SST interneuron signaling" needs to be confirmed by excluding potential changes in firing rate and synaptic transmission over time during optogenetic stimulation. Figures displaying firing rates and IPSC amplitude over time need to be shown.

See our response to point 1 and 2.

6. Page 5 line 118, authors conclude: "Altogether, these data demonstrate that astrocytes modulate their Ca²⁺ dynamics coherently with the previous history of interneuron network activity unveiling in these non-neuronal cells a memory mechanism of the past events which occurred in the GABAergic interneuron networks." This a very important conclusion that therefore requires the proper support by excluding changes in GABAergic transmission, as specifically indicated in points 1 and 5.

7. Supplementary Fig 4 is unclear; e.g., which IPSC is measured, the first or the last IPSC in the train? The important data is actually not shown, and plots of the individual IPSC amplitude over time (for both the first and second train of stimuli) need to be shown. Changes due to short term facilitation or depression need to be checked.

The results reported in Suppl. Fig. 11 indicate that both the firing rate of PV and SST interneurons and IPSC amplitude in pyramidal neurons are unchanged during the successive optogenetic stimulations.

Also, while the authors measured IPSC charge, the analysis of the IPSC peak amplitude seems more appropriate because this parameter would more directly reflect the amount of GABA released (the charge may be affected by the activity of GABA transporters).

We measured and reported the IPSC peak amplitudes as suggested.

Minor comments

1. There are several acronyms and terminology used throughout the text that is not defined or unclear. For example, Page 3, line 65, what is vxc? On Page 7, line 159, define G42.

We corrected the typo and in the Methods we defined the G42 mouse line.

2. When referencing GABAergic signaling into astrocytes in page 6, line 149, additional original references on this issue should be included: Navarrete and Araque 2008; Gould et al. Philos Trans R Soc Lond B 2014; Perea et al. eLife 2016.

3. Page 10, line 252, when discussing Ca²⁺ signal regulation evoked by neuronal signals, the references Perea and Araque, J Neurosci 2005 and Schipke et al, Cer Cortex 2008 may be included.

We included the suggested references.

Reviewer #3 (Remarks to the Author):

We thank the reviewer for the appreciation of our work.

According to the reviewer's suggestion, we extended the discussion on the potential functional role of GABAergic signaling to astrocytes. We also applied a color LUT to the movies that, indeed, enhanced the visibility of the Ca²⁺ elevations.

Reviewer #4 (Remarks to the Author):

We thank the reviewer for the comment on our study that *"shows very interesting data suggesting cell-specific interneuronal-astrocytic networks"*. We also thank the reviewer for his/her constructive comments and suggestions that have significantly improved the manuscript.

Major concerns:

1) There is a frequent absence of direct evidence of GABAergic actions onto astrocytes. All the results rely on the GABA released from interneurons activates GABA_B receptors in astrocytes. However, because the experiments were done in absence of blockage of other neurotransmitter receptors (i.e., glutamate), the particular connectivity of PV- SOM would result in a disinhibition of the local inhibitory network, which might cause the boosting calcium responses of astrocytes derived from glutamatergic signaling. Thus, indirect effect of GABA through neuronal network would underlie the enhanced Ca signals in astrocytes after 10 and 30 s stimulation protocols.

This is a good point that was also raised by another reviewer. As suggested, we performed additional experiments in the presence of blockers of excitation, (NBQX and AP5 for AMPA and NMDARs, respectively, and MPEP for type 5 metabotropic glutamate receptors). We found that under these conditions the enhanced Ca²⁺ response of GCaMP6f-expressing astrocytes to 10 and 30 light pulse stimulation of SST interneurons was unaffected. These data are reported in the new Suppl. Fig. 10 of the revised manuscript.

2) Authors show IPSC charge after light stimulation; however, it's unclear how they measured it. Considering that changes in astrocyte Ca signals happen after 10-20s after interneuron stimulation, it would be necessary to test the time course of GABA levels (IPSCs) to ensure no changes in the inhibitory synaptic transmission.

In the Supplementary Fig. 4 of the original manuscript, we reported the results from control experiments that evaluated the possible changes of the interneuron response to the successive sequences of light pulses as well as of the evoked IPSCs in pyramidal neurons. These experiments aimed to clarify whether a change in the synaptic release of GABA could account for the decreased and the enhanced astrocyte response to PV and SST interneuron activation, respectively.

We see how the original Supplementary Fig. 4 reporting these data was unclear (the same figure was also similarly criticized by another reviewer). Following this criticism, in the new Suppl. Fig. 11 we now report the action potential firing rate of PV and SST interneurons as well as the amplitude of the IPSC in each light pulse of the two sequences of 10 and the two sequences of 30 light pulse stimulation. The results of these control experiments suggest that the different astrocyte response to PV and SST interneurons is unlikely due either to a change in the responsiveness of PV and SST interneurons to the successive optogenetic light pulse stimulations or to a change in evoked synaptic GABA release.

3) There are some confusing data regarding the recording and analysis of cell soma Ca signals. In vivo experiments shown in Figure 1 display astrocyte Ca increases at the soma after PV and Som interneuron stimulation, but no statistically significant, Figure 2. "Based on the sparse nature of GCaMP6f expression in astrocytes", authors change the approach and use Fluo-4 loaded astrocytes to record somatic Ca signals, and then they found a robust effect for both PV and Som interneurons, Figure 3, 4. However, they use again the GCaMP6f expression to monitor the effects of somatostatin peptide on the Ca responses in microdomains, proximal

processes and cellular soma, Figure 6. In that case, the found robust responses in all of them. Therefore, what is the final message? Do PV and Som interneurons induce Ca signals at the astrocytic soma? The information shown in Figure 2, which is the average of *in vivo* and slice data, has to be revised as well as the message of the text.

We understand the point raised by the reviewer. In our opinion, the lack of a significant increase in the Ca²⁺ response at the soma of astrocytes reported in the original Fig. 2, could be due to the relatively low number of astrocytes recorded in these experiments: 20 GCaMP6f-expressing astrocytes (11 from *in vivo* and 9 from slice preparations) in Chr2-PV mice; 21 GCaMP6f-expressing astrocytes (10 from *in vivo* and 11 from slice preparations) in Chr2-SST mice. Consistent with this hypothesis, when a larger number of GCaMP6f-expressing astrocytes (n = 57) were analyzed, as in the case of Fig. 6, the somatic Ca²⁺ responses resulted to be significantly increase upon SST interneuron stimulation.

Please also note that to provide further evidence that somatic Ca²⁺ elevations occur in response to interneurons *in vivo*, we performed additional experiments that after astrocyte loading with the Ca²⁺ indicator Oregon Green BAPTA-1 allowed us to follow the somatic Ca²⁺ signal from a large number of *in vivo* astrocytes (100 astrocytes in Chr2-PV mice and 90 astrocytes in Chr2-SST mice). The results reported in the new Fig. 3, confirmed a statistically significant increase of somatic Ca²⁺ elevations in OGB1-loaded astrocytes in response to both PV and SST interneuron activation.

4) As authors indicated in the abstract interneurons “differentially signal to astrocytes at fine processes weak and robust GABAB receptor-mediated Ca elevations”; however, they did not test the dependence of local Ca signals to GABA_B receptor activation. They analyzed somatic Ca signal induced by intense protocols of stimulation of interneurons (30); but, were microdomains sensitive to SCH? Considering the importance of microdomains as key points in close contact with synapses, it would be interesting to elucidate whether GABA_B antagonist can abolish the observed responses by moderate (10) and stronger (30) stimuli.

As we now report in the new Suppl. Fig. 7, the increased Ca²⁺ elevations at different astrocytic compartments, including microdomains, in response to interneuron signalling were blocked in the presence of the specific GABA_BR antagonist SCH50911.

5) Figure 6a: the actual data cannot exclude the effects of the neuropeptide Somastotatin on neuronal network. To ensure that somatostatin peptide induces Ca signals in astrocytes by activation of SSTR4 receptors a cocktail of antagonists for different neurotransmitter receptors (plus TTX) should be used.

We agree with the reviewer that it is important to exclude the possibility that the somatostatin peptide has network effects that may indirectly affect astrocytic Ca²⁺ signals. Accordingly, we performed new experiments to evaluate the astrocyte response to SST interneuron signaling in the presence of both TTX and different receptor antagonists, including MPEP (50 μM, for metabotropic GluR5), PPADS (100 μM, for purinergic P2 receptors) and SCH50911 (40 μM, for GABA_B receptors). Obtained results reveal that after blocking glutamatergic, purinergic and GABAergic metabotropic receptors the astrocyte response to SST interneurons remained unchanged. These results are included in the new Fig. 6.

6) The modulatory role of somatostatin peptide in the astrocyte Ca signal is an interesting result. If so, would somatostatin peptide boost the Ca responses to local application of GABA? Could PV stimulation plus somatostatin peptide (in the bath) mimic the astrocyte responses to Som interneurons?

We performed the experiments suggested by the reviewer. Results obtained revealed that the second episode of 30 pulse PV interneuron stimulation performed in the presence of a bath perfusion with the neuropeptide Somatostatin (2 μ M), failed to induce the depression of the astrocyte response otherwise observed in the absence of Somatostatin. with respect to the response induced by the first 30 pulse stimulation. The response was, however, not potentiated indicating that the addition of the Somatostatin peptide mimicked only in part the astrocyte response to SST interneurons. These data are reported in the new Fig. 7a,b. It seems to us that several experimental caveats justify the partial effect observed following exogenous somatostatin applications. The bath perfusion of the peptide can hardly mimic the spatial-temporal features of the somatostatin release from synaptic terminals. Due to the poor penetration of peptides into brain tissue, it is also probable that the peptide concentration reached at the proper sites are insufficient to exert a full effect on the astrocytes response.

7) Although the idea is tempting, there is no evidence from the actual data to support the author's conclusion of the existence of a cellular memory mechanism in astrocytes. -Proper controls of neuronal network activity during and after PV and Som stim are required to discard that the increase of spontaneous Ca oscillations in astrocytes is derived from network activity.

-What are the mechanisms regulating this memory?

-Do astrocytes modulate their GABA_B receptor or GABA transporter levels in the membrane as response to the history of GABAergic network activity?

The results from the new experiments that are reported in the revised version of the manuscript (see Suppl. Fig. 11) provide further support to the view that the change in the astrocytes response to GABAergic interneurons is unlikely due to a change in synaptic GABA release or to a networks effects and it may rather represent an intrinsic astrocytic property. We also provide evidence for the absence of significant contribution of glutamatergic signalling to the astrocyte response evoked by interneuron activity (see Suppl. Fig. 10). Nevertheless, we cannot but agree with the reviewer entirely that to conclusively support the existence of a cellular memory form in astrocytes, it will be necessary to identify the molecular mechanism. Therefore, we toned down our comments about this issue. We believe, however, that our data are important as they prompt an investigation addressing whether a non-neuronal element of brain circuits, such as the astrocyte, is actively involved in memory function.

Minor points:

Figure 6a mentions Ca signals induced by somatostatin peptide, but not by somatostatin interneurons.

We corrected the typo

The analysis of response ratio (RR) shown in Figure 7 is confusing. Why authors put together signals coming from microdomains and proximal processes that showed different behavior to interneuron stimulation?

From the methods is unclear to me whether authors summarized values of frequency and amplitude from those structures in a single data point. If so, why?

We put together the signals coming from proximal and fine processes because these are presumed main targets of synaptic neurotransmitters. As regards the question about the frequency and the amplitude, we considered that the plasticity of a specific cellular event, such as the astrocytic Ca²⁺ transient evoked by a given stimulus, can be expressed as a change in either the frequency, the amplitude or both the frequency and the amplitude (see Araque et

al., 2014). We therefore put together these two parameters to obtain an index of the response and then evaluate how this index eventually changes following the successive interneuron stimulations.

Reviewers' comments:

Reviewer #1 (Remarks to the Author):

I appreciate that the authors put significant effort into the revision process. However, several problems remain, which from my point of view preclude publication in a high profile journal.

Major points:

1. Regarding my previous point 2, I still would like to see a plot of amplitude of Ca²⁺ transients against time on an uninterrupted time axis. Plotting "imaging acquisitions" is insufficient.
2. Regarding my previous point 3, although the authors have substantially attenuated their provocative statements, the problem remains that the physiological significance of the observed effects is unclear. Furthermore, it remains open how long the reported plasticity lasts.
3. Regarding my previous point 4, my criticism regarding the recording in anesthetized animals still stands. The authors now state that Ca²⁺ signaling in astrocytes is affected by anesthetics, but that only reinforces my point. They should perform Ca²⁺ imaging experiments in awake behaving animals with and without interneuron stimulation to clarify the issue.
4. Regarding my previous point 6, the authors now report the negative result that HC-030031 fails to affect the Ca²⁺ transients. However, the question of the molecular mechanisms of the Ca²⁺ transients remains open. Furthermore, how can the authors be sure that the concentration was sufficient?
5. Playing with SST receptor agonists and somatostatin to convert facilitating into depressing responses and vice versa may be interesting for specialized readers, but tells us very little about the physiological significance of these effects.
6. The authors have largely ignored my comments on the specificity of the antibodies. As we all know, pre-absorption controls are insufficient. Rather, knockout controls are required. I looked extensively, but I cannot find any such data in the literature for these antibodies. Furthermore, the anti-SSTR4 antibody is not on the Santa Cruz Biotechnology webpage anymore (with many antibodies discontinued by this company). If the authors want to maintain figure 5, these issues need to be clarified.

Minor points:

p. 5, center: A clear definition of microdomain is never provided. Probably, it would be better to talk about proximal and distal processes and specify a criterion distance for separation.

p. 6, center: The sustained Ca²⁺ response in the presence of synaptic blockers is a key

finding and accordingly has to be incorporated into the main body of the manuscript.

p. 6, bottom: The time interval between the two stimulations needs to be repeated in the results section.

p. 16, bottom: Apparently all recordings were done at room temperature. Control experiments at physiological temperature need to be provided.

Legends throughout: The concentrations of all chemicals need to be specified in the legends. Furthermore, it is still not clear enough which experiments were performed in vitro and which in vivo.

Reviewer #2 (Remarks to the Author):

Authors have adequately addressed the concerns expressed about the original version. The most important concern was related to the potential changes in GABA release evoked by successive stimuli. This possibility has been elegantly ruled out by showing that postsynaptic responses evoked by two successive stimuli were identical. I believe this is an important piece of information that should be reported as a main Figure in the main text rather than a supplementary figure.

It is also interesting that PV-IPSC amplitude decayed during the 10 or 30Hz stimulus, whereas SST-IPSCs were rather constant. However, authors should comment these differences because they may underlie the differential effects on signaling from both types on interneurons. For instance, if PV were stimulated with a ramp of increasing frequency stimulation to maintain a constant IPSC amplitude during the stimulus train, like it occurs in SST interneurons, would the astrocyte responsiveness be facilitated by successive trains, as in SST-evoked signaling? I think this type of experiment is not necessary for the present work, but it would add valuable information, and authors may want to test it to strengthen the properties of astrocyte-interneuron signaling. Nevertheless, I understand this question might be out of the scope of the present work, which focuses on changes in astrocyte responses by successive stimuli.

Since all the concerns have been adequately addressed and the conclusions are now fully supported, I reaffirm my original assessment of the manuscript:

This is an important, clear and well-performed study that presents novel and interesting results. It adds valuable information regarding astrocyte-neuron interaction, a relevant and emerging, yet debated, topic in neuroscience.

Reviewer #3 (Remarks to the Author):

Mariotti and colleagues submitted a revised version of their manuscript. The work describes

a novel pathway of neuron-glia communication. By using a set of state-of-the-art techniques they demonstrate that a distinct population of interneurons (SST neurons) do not only stimulate astroglial calcium signals via activation of GABAB receptors on astrocytes, but also they show that co-release of somatostatin (SST) can significantly potentiate the glial response, in contrast to SST-lacking parvalbumin-positive interneurons. Thereby, this work describes a novel molecular pathway how different cell types cooperate for proper brain function.

The authors addressed my comments for the Discussion and changed the LUT of their movies as requested.

Reviewer #4 (Remarks to the Author):

Authors have reviewed the previous comments and revised the manuscript accordingly.

However, some issues have been detected in this version that should be considered:

1) Suppl. Fig 6: Example shown in figure 6d showing astrocyte Ca signals from SST mice do not represent the population behavior. Change accordingly to show changes at the level of the soma as indicated in panel 6f.

2) Suppl. Fig 11: Although it's clear that firing rate of PV and SST cells did not change during light protocol stimulation, for the case of PV cells and IPSC currents it seems that IPSCs amplitude decreased during the stim protocol. Required quantification needs to be shown in this figure to undoubtedly discard a reduction in GABA released by PV interneurons.

3) In order to clarify the message, authors should reconcile the data observed in figure 2 and 3. They were obtained using different approach to record Ca signals, but the conclusion of those experiments is that somatic Ca signals are modulated by longer stimulation protocols of PV and SST neurons. Thus, data shown in Figure 3 should be shown as additional panels of Figure 2.

Point-by-point response to the reviewers

Reviewer #1

Major points:

1. Regarding my previous point 2, I still would like to see a plot of amplitude of Ca²⁺ transients against time on an uninterrupted time axis. Plotting "imaging acquisitions" is insufficient.

As requested by the reviewer, we now report for each 150-second imaging session, the amplitude of Ca²⁺ transients on an uninterrupted time axis in *in vivo* (Supplementary Figure 12a) and SSCx slice (Supplementary Figure 12d) experiments.

2. Regarding my previous point 3, although the authors have substantially attenuated their provocative statements, the problem remains that the physiological significance of the observed effects is unclear. Furthermore, it remains open how long the reported plasticity lasts.

The data presented in our manuscript provide a new perspective on the functional role of astrocytes as specific targets of GABAergic interneuron signalling in the brain. While we fully understand the request of the reviewer, we believe that the clarification of this role is beyond the scope of this study and it can represent an important goal of future experiments - ideally, in awake behaving animals - not only from our group but also from others.

As regards the reviewer's question on the astrocyte response plasticity, we performed additional experiments. The results obtained reveal that the potentiation of the astrocyte response to successive stimulations of SST interneurons is a transient phenomenon that is "... *hardly detectable with 10-min intervals and absent with 20-min intervals (Supplementary Fig. 11)*" (page 7, lines 158,159).

3. Regarding my previous point 4, my criticism regarding the recording in anesthetized animals still stands. The authors now state that Ca²⁺ signaling in astrocytes is affected by anesthetics, but that only reinforces my point. They should perform Ca²⁺ imaging experiments in awake behaving animals with and without interneuron stimulation to clarify the issue.

We understand the point raised by the reviewer. However, at present we are not equipped yet to properly investigate the calcium signal in astrocytes from awake behaving animals. Setting up these experiments is certainly a major goal in our future research but clearly beyond the scope of the present work.

4. Regarding my previous point 6, the authors now report the negative result that HC-030031 fails to affect the Ca²⁺ transients. However, the question of the molecular mechanisms of the Ca²⁺ transients remains open. Furthermore, how can the authors be sure that the concentration was sufficient?

We agree with the reviewer and on page 13, lines 326,327, we now state that "*The signalling pathways mediating the Ca²⁺ microdomain activity remains to be clarified.*"

As to the concentrations, we used HC-030031 at 80 μM, a concentration that is well above the IC₅₀ (6.2 μM) measured on the Ca²⁺ influx induced in TRPA1-expressing cells by the specific TRPA1 agonist AITC (see McNamara et al., PNAS 2007). The risk with higher concentrations is the loss of action specificity.

5. Playing with SST receptor agonists and somatostatin to convert facilitating into depressing

responses and vice versa may be interesting for specialized readers, but tells us very little about the physiological significance of these effects.

The data the Reviewer is referring to were obtained from experiments that were performed following a specific request of one of the other reviewers. In our opinion, these data strengthen our conclusion on the importance of the neuropeptide Somatostatin in the astrocyte response to SST interneurons and will help to develop specifically designed experiments aimed to understand the functional role of the SST interneuron specific signalling to astrocytes.

6. The authors have largely ignored my comments on the specificity of the antibodies. As we all know, pre-absorption controls are insufficient. Rather, knockout controls are required. I looked extensively, but I cannot find any such data in the literature for these antibodies. Furthermore, the anti-SSTR4 antibody is not on the Santa Cruz Biotechnology webpage anymore (with many antibodies discontinued by this company). If the authors want to maintain figure 5, these issues need to be clarified.

We apologize for not having specified in our previous reply why we could not address the original comment of the reviewer on the specificity of the antibodies. The problem was that we were unable to obtain the specific knockout mice. Thanks to the collaborative effort of Cécile Viollet and Bernhard Bettler, we had then the possibility to perform the requested experiments on SSTR4 and GABA_{B2} knockout mice. The results obtained are reported in the new Supplementary Figs 15, 16 as well as in the new Tables 2, 3 and confirm the specificity of the antibodies that we used.

Furthermore, please note that Santa Cruz was recently involved in a series of legal problems (see links below) and lost their license to use animals. A myriad of polyclonal antibodies were discontinued because of this.

<https://www.nature.com/news/us-government-issues-historic-3-5-million-fine-over-animal-welfare-1.19958>

<http://blogs.sciencemag.org/pipeline/archives/2016/05/23/trouble-at-santa-cruz-biotechnology>

Minor points:

p. 5, center: A clear definition of microdomain is never provided. Probably, it would be better to talk about proximal and distal processes and specify a criterion distance for separation.

We explained how we identified microdomains in the Methods section (page 24, lines 619,620). For the identification of microdomains, proximal processes and soma, we used a procedure adapted from the GECIquant plugin in ImageJ (see Srinivasan et al., 2015). We understand the point of the reviewer and we agree that a satisfactory definition of microdomains is lacking in the field. This is also due to our poor understanding of the Ca²⁺ events that are associated with astrocytic structures, such as the fine processes, that are well below the optical resolution of the currently available microscopes. A new analytical approach of microdomain activity is certainly needed.

Notwithstanding, we would prefer to maintain the term *microdomains* that has been historically associated to Ca²⁺ elevations at the finest astrocytic processes.

The distance from the soma can be alternative good criterium, as suggested by the reviewer, but it would exclude the activity of a number of microdomains that occurs close to the soma. This is why we preferred a different criterium that is not, of course, without limitations.

p. 6, center: The sustained Ca²⁺ response in the presence of synaptic blockers is a key finding and accordingly has to be incorporated into the main body of the manuscript.

Following the suggestion of the reviewer, we now include these data in the new principal Figure 7.

p. 6, bottom: The time interval between the two stimulations needs to be repeated in the results section.

We now report also in the Results section the time interval between the two stimulations.

p. 16, bottom: Apparently all recordings were done at room temperature. Control experiments at physiological temperature need to be provided.

The reviewer is probably referring to recordings in brain slice experiments that were, indeed, done at room temperature. Importantly, however, the main finding of our study, i.e. the different properties of the astrocyte response to GABAergic signalling from PV and SST interneurons, were observed both in brain slice preparations (at room temperature) and in *in vivo* preparations (at physiological temperature). Given the congruence of the results, it seems to us unnecessary to replicate at physiological temperature the results obtained in brain slice experiments.

Legends throughout: The concentrations of all chemicals need to be specified in the legends. Furthermore, it is still not clear enough which experiments were performed in vitro and which in vivo.

We now provide the information requested.

Reviewer #2

We thank the reviewer for stating that "*all the concerns have been adequately addressed and the conclusions are now fully supported.*"

Reviewer #3

We thank the reviewer for the positive comments on our study.

Reviewer #4 (Remarks to the Author).

We thank the reviewer for the constructive comments on our study.

The three additional comments to the revised version were addressed as follows:

1) Suppl. Fig 6: Example shown in figure 6d showing astrocyte Ca signals from SST mice do not represent the population behavior. Change accordingly to show changes at the level of the soma as indicated in panel 6f.

Following the reviewer's comment, we replaced the original astrocyte with a more representative one that also exhibits somatic Ca²⁺ responses to SST interneuron signalling.

2) Suppl. Fig 11: Although it's clear that firing rate of PV and SST cells did not changed during light

protocol stimulation, for the case of PV cells and IPSC currents it seems that IPSCs amplitude decreased during the stim protocol. Required quantification needs to be shown in this figure to undoubtedly discard a reduction in GABA released by PV interneurons.

We understand the point raised by the reviewer. Accordingly, we performed a new quantitative analysis of the reduction in IPSC amplitude that we observed over the course of each stimulation of PV interneurons. Results are reported in the modified Supplementary Fig. 13 and commented on page 8, lines 184-191, as follows:

" In the case of PV interneurons, in the two sets of 30 light pulses, besides an unchanged firing rate, we observed a reduction in IPSC amplitude that can be indicative of a GABA receptor desensitization and/or a decrease in synaptic GABA release. It is noteworthy that the IPSC reduction, in terms of both time course and relative amplitude (Supplementary Fig. 12), was similar during the two PV interneuron stimulations suggesting that a decrease in GABA release cannot explain the impairment of the astrocyte response to the second PV interneuron stimulation. "

3) In order to clarify the message, authors should reconcile the data observed in figure 2 and 3. They were obtained using different approach to record Ca signals, but the conclusion of those experiments is that somatic Ca signals are modulated by longer stimulation protocols of PV and SST neurons. Thus, data shown in Figure 3 should be shown as additional panels of Figure 2.

We followed the suggestion of the reviewer and included in Figure 2 the quantitative data on somatic Ca^{2+} signals that were originally reported in Figure 3. Please note that, due to space constraints, panels **a** and **b** of the original Figure 3 could not be included in this figure and were moved to Supplementary Fig. 8.

REVIEWERS' COMMENTS:

Reviewer #1 (Remarks to the Author):

(1) The data should be presented more quantitatively at several places. On p. 6, bottom, the P values should be given. On p. 7, top, quantification is needed. On p. 8, top, "was similar" should be supported by statistics. On p. 10, center, "significant depression" needs to be again supported by quantitative statements. Fill words like "It is noteworthy" should be deleted.

(2) A clear definition of microdomain is still lacking.

(3) The authors claim that a decrease or increase in GABA release cannot explain the dynamic change in the astrocytic responses (p. 8, top), but the evidence for this is at best circumstantial.

(4) Relatedly, the authors interpret their findings in the context of somatostatin receptor expression in postsynaptic astrocytes. However, somatostatin receptors in neurons (including axons and presynaptic terminals) may also contribute to the observed effects. These alternative interpretations need to be considered more carefully.

It should be in the interest of the authors to rectify these problems.

Reviewer #4 (Remarks to the Author):

All the comments have been addressed and properly clarified.

Comment 2: authors mentioned Supp. Fig. 12, but it's actually Supp. Fig. 13.

Point -by-point response to Reviewer 1

(1) The data should be presented more quantitatively at several places. On p. 6, bottom, the P values should be given. On p. 7, top, quantification is needed. On p. 8, top, "was similar" should be supported by statistics. On p. 10, center, "significant depression" needs to be again supported by quantitative statements. Fill words like "It is noteworthy" should be deleted.

We addressed all these specific requests as follows:

- On original p. 6, we include the p value (**p = 0.949**) related to the delay of the astrocyte response to 10 and 30 light pulse activation of SST interneurons.
- On p. 7, we have modified the text as follows: "*We found that the potentiation is a transient phenomenon as it was absent with 20-min intervals and observed with 10-min intervals **only as a small, albeit significant (p = 0.031), increase in the mean number of active microdomains (Supplementary Fig. 11).***"
- On p.8, we now include the p values that support our original statement. The sentence is now as follows: "*The IPSC reduction during the two sets of 30 pulse PV interneuron stimulation was, however, similar, in terms of both its time course (p = 0.66 and p = 0.83 for the fast and the slow time decay component, respectively) and amplitude (p = 0.55; Supplementary Fig. 13 e,f).*"
- On p 10, we now provide a more quantitative evaluation of the "significant depression" and also include exact p values. The text has been modified as follows: "*... in the presence of CYN154806, the astrocyte Ca²⁺ response ... **exhibited a significant reduction in the mean number of microdomains and the mean event frequency (p = 0.007 and p = 0.003, respectively; Fig. 6d,e, lower panels)***".

Sentences like "it is noteworthy" have been deleted throughout the text.

(2) A clear definition of microdomain is still lacking.

In the Results section, we now state that "*We first studied Ca²⁺ signal dynamics in different compartments of GCaMP6f-expressing astrocytes including the soma, the proximal processes*

and the fine processes exhibiting spatially restricted Ca²⁺ transients, i.e. Ca²⁺ microdomains, ...".

(3) The authors claim that a decrease or increase in GABA release cannot explain the dynamic change in the astrocytic responses (p. 8, top), but the evidence for this is at best circumstantial.

We now tone down our claim and on the original p. 8, top, we add the following sentence "*However, direct measurements of GABA concentrations would be necessary to validate this conclusion*".

(4) Relatedly, the authors interpret their findings in the context of somatostatin receptor expression in postsynaptic astrocytes. However, somatostatin receptors in neurons (including axons and presynaptic terminals) may also contribute to the observed effects. These alternative interpretations need to be considered more carefully.

We now discuss this issue in the Discussion as follows: "*Different SSTRs are also expressed on neuronal axon terminals and they are proposed to cooperate with presynaptic GABARs in the control of glutamate release. We cannot, therefore, exclude that a change in the network activity mediated by neuronal SSTR activation may contribute to modulate the astrocyte response to SST interneuron signalling.*"